# Oxidised metabolites of the omega-6 fatty acid linoleic acid activate dFOXO

So Yeon Kwon[1,]*, Karen Massey[2,]*, Mark A Watson[1,]*, Tayab Hussain[1], Giacomo Volpe[1], Christopher D Buckley[3,4], Anna Nicolaou[2,5], Paul Badenhorst[1]

**Obesity-induced inflammation, or meta-inflammation, plays key roles in metabolic syndrome and is a significant risk factor in diabetes and cardiovascular disease. To investigate causal links between obesity, meta-inflammation, and insulin signaling we established a *Drosophila* model to determine how elevated dietary fat and changes in the levels and balance of saturated fatty acids (SFAs) and polyunsaturated fatty acids (PUFAs) influence inflammation. We observe negligible effect of saturated fatty acid on inflammation but marked enhancement or suppression by omega-6 and omega-3 PUFAs, respectively. Using combined lipidomic and genetic analysis, we show omega-6 PUFA enhances meta-inflammation by producing linoleic acid–derived lipid mediator 9-hydroxy-octadecadienoic acid (9-HODE). Transcriptome analysis reveals 9-HODE functions by regulating FOXO family transcription factors. We show 9-HODE activates JNK, triggering FOXO nuclear localisation and chromatin binding. FOXO TFs are important transducers of the insulin signaling pathway that are normally downregulated by insulin. By activating FOXO, 9-HODE could antagonise insulin signaling providing a molecular conduit linking changes in dietary fatty acid balance, meta-inflammation, and insulin resistance.**

## Introduction

Obesity is a growing global public health challenge with 650 million adults worldwide estimated to be obese (WHO, 2016). The accompanying changes in the levels and balance of fatty acids are key components of metabolic syndrome, a multiplex risk factor underlying cardiovascular disease that includes as cofactors insulin resistance and chronic inflammatory states (Grundy, 2016). In turn, both insulin resistance and meta-inflammation—a chronic, low-grade inflammatory response in metabolic tissues—are speculated to be consequences of the increased uptake of dietary fatty acids and changes in the dietary balance of fatty acids that accompany obesity (Gregor & Hotamisligil, 2011; Shulman, 2014). Meta-inflammation is characterized by elevated inflammatory cytokine levels (Hotamisligil et al, 1993), enhanced proinflammatory signaling, and macrophage activation (Osborn & Olefsky, 2012). Meta-inflammation in combination with changes in fatty acid flux has been proposed to antagonise insulin signaling leading to insulin resistance (Yin et al, 1998; Yuan et al, 2001; Szendroedi et al, 2014). Under normal conditions, engagement of the conserved insulin/insulin-like growth factor signaling (IIS) pathway leads to Akt activation and the phosphorylation of FOXO (Forkhead box O) transcription factors (TFs). Phosphorylation of FOXO by Akt antagonizes FOXO function by controlling nuclear-cytoplasmic distribution and is the ultimate effector of IIS, regulating not only glucose homeostasis but also growth and longevity (Taguchi & White, 2008). Identifying how meta-inflammation and altered dietary fatty acids impact the IIS pathway are key to unraveling the mechanisms underlying the development of insulin resistance.

Previously, it has been shown that interventions that inhibit inflammatory pathways are able to prevent insulin resistance that accompanies metabolic syndrome (Yin et al, 1998; Yuan et al, 2001). As such control of inflammation presents an attractive target for therapeutic intervention. One potential route by which inflammation could be influenced is through changes in the dietary levels of fatty acids, in particular the relative ratio of omega-3 to omega-6 polyunsaturated fatty acid (PUFA). The link between omega-3 and omega-6 PUFAs and inflammation is well documented with fatty acids from both families serving as precursors for the production of immunomodulatory, oxygenated metabolites (Fritsche, 2006). PUFAs of

[1]Institute of Cancer and Genomic Sciences, University of Birmingham, Edgbaston, UK    [2]Bradford School of Pharmacy, University of Bradford, Bradford, UK    [3]Institute of Inflammation and Ageing, Centre for Translational Inflammation Research, Queen Elizabeth Hospital, Edgbaston, UK    [4]Kennedy Institute of Rheumatology, University of Oxford, Oxford, UK    [5]Laboratory for Lipidomics and Lipid Biology, Division of Pharmacy and Optometry, Faculty of Biology, Medicine and Health, Manchester Academic Health Science Centre, Manchester, UK

Correspondence: p.w.badenhorst@bham.ac.uk
Karen Massey's present address is BASF Pharma, Isle of Lewis, UK
Mark A Watson's present address is Buck Institute for Research on Aging, Novato, CA, USA
Giacomo Volpe's present address is Key Laboratory of Regenerative Biology, Joint School of Life Sciences, Guangzhou Institutes of Biomedicine and Health, Guangzhou, China
*So Yeon Kwon, Karen Massey, and Mark A Watson are co-first authors

the omega-6 family are broadly considered proinflammatory, whereas omega-3 PUFAs have anti-inflammatory actions (Simopoulos, 2002; Serhan et al, 2008). In humans, proinflammatory effects of omega-6 PUFAs are attributed to the production of eicosanoid (20 carbon chain [C20]) lipid mediators such as prostaglandins derived from arachidonic acid (AA) via cyclooxygenase (COX), lipoxygenase (LOX), and cytochrome P450 (CYP) enzymes. Production of such omega-6 PUFA-derived eicosanoids has been shown to be required for insulin resistance (Li et al, 2015), whereas, conversely, increasing levels of anti-inflammatory omega-3 PUFA-derived lipid mediators appear to ameliorate insulin resistance (Yin et al, 1998; Yuan et al, 2001; Oh et al, 2010). Taken together, these data suggest that changing the spectrum of downstream lipid mediators by reducing dietary levels of omega-6 PUFA or altering their ratio to anti-inflammatory omega-3 PUFA could offer a facile route to control meta-inflammation and insulin resistance.

The predominant omega-6 PUFA in the diet is the 18 carbon (C18) omega-6 PUFA linoleic acid (LA) (Ervin et al, 2004). Although LA can serve as the precursor of AA, indirectly producing prostaglandins and other eicosoanoid lipid mediators, LA is also a direct target of COX, LOX, or myeloperoxidase (MPO) enzymes, generating the octadecanoid (C18) lipid mediators 9- and 13-hydroxy-octadecadienoic acid (HODE) (Kaduce et al, 1989; Laneuville et al, 1995; Zhang et al, 2002; Kubala et al, 2010). 9-HODE modulates leukocyte migration, regulates inflammation by altering expression of proinflammatory cytokines, and is elevated in inflammatory conditions (Moch et al, 1990; Henricks et al, 1991; Jira et al, 1998; Waddington et al, 2001; Mabalirajan et al, 2013). However, little is known of the functions of LA-derived metabolites such as 9-HODE in meta-inflammation and whether they play a role in the control of the IIS pathway. To explore these questions, we have developed a *Drosophila* model system that allows the function of PUFA-derived mediators to be investigated.

*Drosophila* possess a highly effective innate immune system containing circulating leukocyte-like cells called hemocytes that provide a genetically tractable model system to dissect innate immune function and inflammation. Three blood cell types occur: macrophage-like plasmatocytes, crystal cells, and lamellocytes (Badenhorst, 2014). Lamellocytes are normally rare but, in response to inflammatory stimuli, differentiate from plasmatocytes and aggregate to produce nodules called "melanotic tumours" (Stofanko et al, 2010). Lamellocyte differentiation is regulated by inflammatory signaling pathways, including the JAK/STAT and Jun N-terminal kinase (JNK) pathways (Zettervall et al, 2004; Tokusumi et al, 2009a, 2009b), with *Drosophila* c-Jun and FOXO TFs activated during lamellocyte differentiation (Tokusumi et al, 2017). Lamellocyte differentiation and melanotic tumour formation provide a convenient readout of inflammation and have been used to screen for modulators of inflammatory signaling pathways (Minakhina & Steward, 2006; Avet-Rochex et al, 2010).

Here, we use this *Drosophila* model to investigate how changes in the balance of dietary fatty acids modulate inflammation. Our analysis defines omega-6 and omega-3 PUFAs as key determinants of inflammation. Using combined gas chromatography and liquid chromatography tandem-mass spectrometry (LC-MS/MS) analysis, as well as genetic and genomic approaches, we identify omega-6 PUFA LA-derived 9-HODE as an important lipid mediator that links changes in dietary fatty acid balance and control of inflammatory signaling to regulation of the IIS pathway.

# Results

## PUFAs modulate *Drosophila* inflammatory responses

To interrogate how dietary fat modulates inflammation, we developed a fly inflammation model based on the constitutively active *Drosophila* JAK mutant strain *hop^Tum^*. This mutation encodes a temperature-sensitive, constitutively active variant of the sole *Drosophila* JAK (*hopscotch* [*hop*]) that drives unregulated STAT activation (Luo et al, 1997) (Fig 1A). The JAK/STAT signaling pathway is a key transducer of inflammatory signals in mammals, where the hallmarks of inflammation include proliferation, activation, and local accumulation of leukocytes and the increased expression of cytokines and inflammation markers. These cognates of inflammation are recapitulated in *hop^Tum^* mutants where ectopic STAT activation drives hemocyte proliferation (Luo et al, 1997), the activation of plasmatocytes to generate lamellocytes (Kwon et al, 2008; Stofanko et al, 2010), and the aggregation and local accumulation of lamellocytes in melanotic tumours (Fig 1B). *hop^Tum^* mutants also display elevated expression of a panel of known markers of *Drosophila* inflammation and putative cytokines such as Upd3 (Fig 1C). As such, activation of the JAK/STAT pathway in *hop^Tum^* mutants initiates an inflammatory response, one consequence of which is the formation of melanotic tumours. However, propagation or resolution of this response can be modulated by intersecting pro- and anti-inflammatory lipid mediators, allowing melanotic tumour incidence in *hop^Tum^* mutants to be used as a simple assay to identify lipid mediators and quantify the effect of dietary precursors on inflammation.

We first used this model to examine how changes in *Drosophila* dietary levels of saturated fatty acids (SFA) or omega-6 and omega-3 PUFAs affected inflammation. *hop^Tum^* flies raised on increasing concentrations of SFA showed no change in tumour incidence, suggesting minimal additive effects of SFAs on *Drosophila* inflammation (Fig 1D). However, dietary supplementation with the omega-6 PUFA LA or omega-3 PUFA α-linoleic acid (ALA) significantly altered melanotic tumour incidence, with the LA enhancing and ALA suppressing inflammation. Thus, at omega-6 PUFA LA concentrations between 0.1% and 0.7%, tumour incidence was significantly increased compared with animals raised on control medium (Fig 1E). LA levels higher than 1% resulted in increased lethality at larval stages and failure to recover appreciable numbers of viable adults (data not shown).

In contrast, increasing media concentrations of ALA reduced overall tumour incidence by 43.8% (Fig 1F). Strikingly, ALA supplementation was able to override immune-stimulatory effects of LA supplementation. When the standard medium was supplemented with both LA and ALA, tumour incidence was still significantly suppressed, even when ALA levels were sub-stoichiometric (Fig 1F). Taken together, these suggest that the dietary balance of omega-6 to omega-3 PUFA, represented here as LA/ALA ratio, is a key modulator of inflammation with omega-6 and omega-3 PUFAs exerting contrasting effects.

## Mediator lipidomics demonstrates *Drosophila* do not form prostaglandins

In vertebrates, control of inflammation by omega-6 and omega-3 PUFAs is achieved through the production of oxidised lipid mediators,

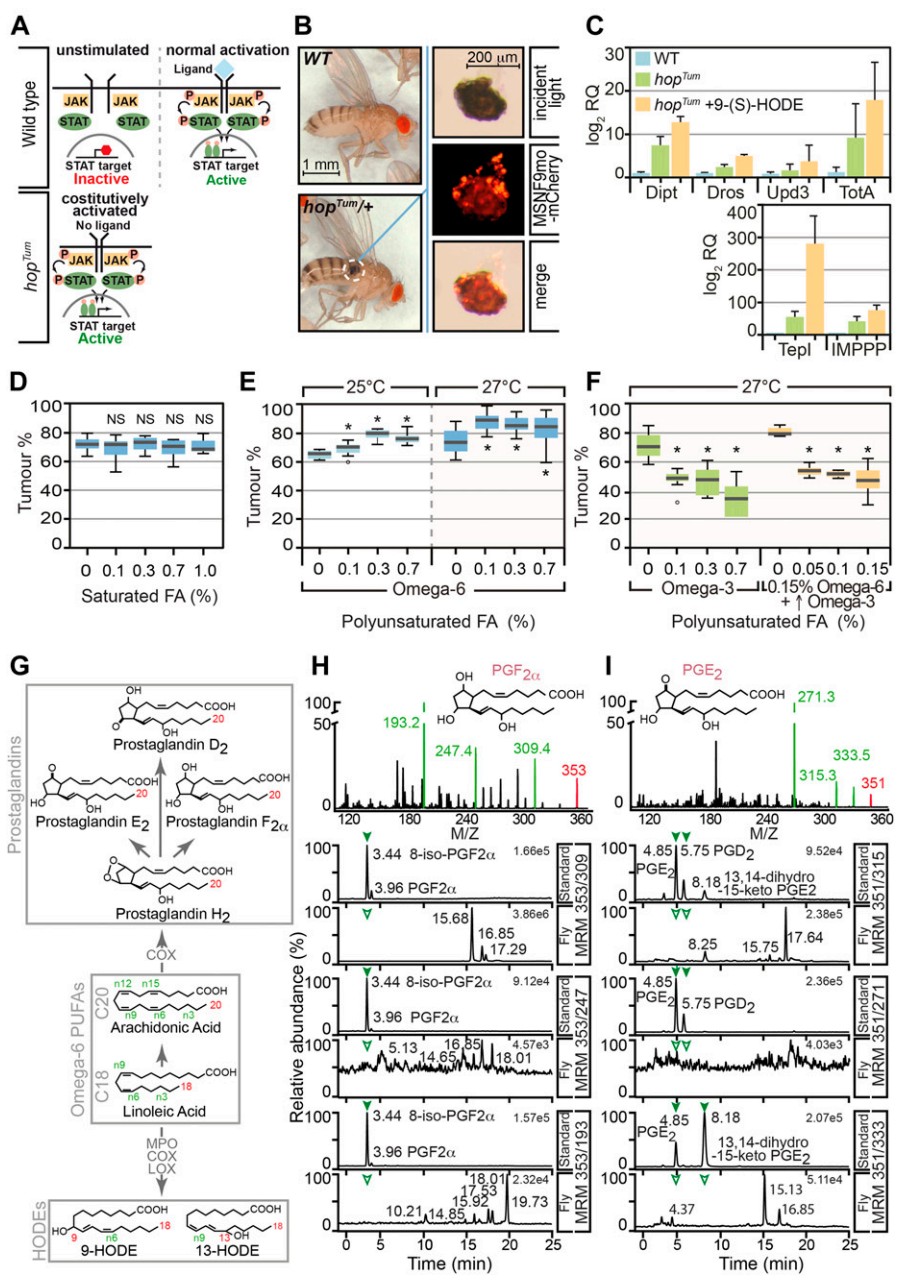

**Figure 1. Immunomodulatory effects of PUFAs on *Drosophila* inflammatory tumours.**
**(A)** Inflammatory melanotic tumour model relies on temperature conditional activation of JAK/STAT in *hop^Tum* gain-of-function heterozygous females. **(B)** Melanotic tumour indicated (dashed circle). Dissected melanotic tumour from *hop^Tum* female carrying the lamellocyte marker (MSNF9-mCherry [Tokusumi et al, 2009a, 2009b]) reveals melanotic tumour is composed of lamellocyte aggregates. **(C)** Real-time RT-PCR was used to determine levels of inflammation markers in control and *hop^Tum* larvae raised in both standard media and media supplemented with 9-(S)-HODE (1 μM). *RpL32* provided an endogenous control to normalise expression. **(D, E, F)** Melanotic tumour incidence was assayed in *hop^Tum* heterozygous females raised on standard medium supplemented with fatty acids. **(D)** SFA supplementation (coconut oil) did not increase tumour incidence. **(E)** Omega-6 PUFA linoleic acid (LA) supplementation increased tumour incidence. **(F)** Omega-3 PUFA (α-linoleic acid [ALA]) supplementation suppressed melanotic tumours even when 0.15% of LA was present. Cultures were incubated at 27°C except for omega-6 PUFA, which was conducted at both 25°C and 27°C. As *hop^Tum* is temperature-sensitive-conditional, changes in culture temperatures modify basal melanotic tumour incidence, allowing better resolution of either suppressors or enhancers of inflammation (Badenhorst et al, 2002). **(G)** Schematic showing biosynthetic pathways of omega-6 PUFA derived lipid mediators. The 18-carbon (C18) omega-6 PUFA LA can be converted to arachidonic acid, the precursor of eicosanoids, a family of potent bioactive lipids that include the cyclooxygenase (COX)-derived prostaglandins. **(H, I)** *hop^Tum* larvae were raised on media supplemented with 0.15% LA and the lipid extract was analysed by LC-MS/MS for the presence of PGF$_{2\alpha}$, PGE$_2$, and PGD$_2$. **(H, I)** Neither PGF$_{2\alpha}$ (multiple reaction monitoring [MRM]: m/z: m/z 353>309, 353>247 and 353>193) nor (I) PGE$_2$ and PGD$_2$ (MRM: m/z 351>315, 351>271, and 351>333) were detected in *Drosophila* lipid extracts. Data information: in (D, E, F), 10 replicate crosses were used for each assay point. Box and whiskers plots were generated using R. * indicates values statistically significantly different from unsupplemented, *P*-value < 0.001 determined using *t* test. NS indicates not statistically significantly different.

including the AA derived prostaglandins (Fig 1G). To determine the mechanism by which omega-6 PUFAs enhance fly inflammation, we first investigated whether *Drosophila* produce prostaglandins. There are previous reports suggesting that prostaglandins occur in flies (Pages et al, 1986). However, screening *Drosophila* larval lipid extracts using liquid chromatography coupled to tandem mass spectrometry (LC-MS/MS) failed to detect any of the primary AA-derived prostaglandins—PGE$_2$, PGF$_{2\alpha}$, and PGD$_2$—in extracts from third instar larvae raised on standard media or media supplemented with omega-6 PUFA (0.15% LA, Figs 1H and I and S1A and B).

As an additional control, we analysed the fatty acid composition of *Drosophila* larva, to determine the prevalence of the prostaglandin precursor AA. Fatty acids of chain length greater than 18

carbons (C18), such as AA, eicosapentaenoic acid, and docosahexaenoic acid, were not detected at significant levels (Table S1). However, C18 LA and ALA were both detected at 7% and 0.2% of total fatty acid respectively (Table S1). The absence of significant levels of AA accounts for the lack of detectable levels of prostaglandins, and we conclude that prostaglandins were not responsible for the proinflammatory effects of omega-6 PUFA supplementation.

## Identification of hydroxy-octadecadienoic acid (HODE) in *Drosophila*

The absence of detectable levels of prostaglandins and AA suggests that other lipid mediators were responsible for the proinflammatory

effects of LA observed in our inflammation model. In humans, LA can be oxidised by COX, LOX, or MPO enzymes to produce 9- and 13-hydroxy-octadecadienoic acid (HODE) (Kaduce et al, 1989; Zhang et al, 2002; Kubala et al, 2010) (Fig 1G), with 9-HODE being elevated during inflammation (Moch et al, 1990; Mabalirajan et al, 2013). We, thus, used LC-MS/MS to screen fly larval lipid extracts for HODEs, detecting both 9-HODE (Fig 2A) and 13-HODE (Fig 2B). Under standard conditions, 9-HODE was the predominant species detected (107 ± 45 pg/mg larvae), whereas supplementation with LA increased levels of both 9-HODE and 13-HODE (Fig 2C). ALA supplementation had no effect on levels of either 9-HODE or 13-HODE. Using chiral LC-MS/MS to determine prevalence of HODE enantiomers, we observed that 9-(S)-HODE was the predominant HODE enantiomer detected under all conditions (Fig 2D).

### 9-(S)-HODE modulates inflammatory responses

As 9-(S)-HODE was the predominant enantiomer detected in larval extracts, we tested if 9-(S)-HODE supplementation could enhance inflammatory responses. We observed clear increases in melanotic tumour incidence in $hop^{Tum}$ females after 1 $\mu$M 9-(S)-HODE treatment (Fig 2E). This was also reflected by increased $hop^{Tum}$ male lethality (Fig 2E). A feature of the $hop^{Tum}$ mutation is that males exhibit more severe inflammatory responses than females. The

extent of male lethality increases with severity of inflammatory responses, providing an additional metric to identify enhancers of inflammation. In addition, elevated expression of known markers of *Drosophila* inflammation and cytokines was observed in $hop^{Tum}$ mutants raised on media supplemented with 9-(S)-HODE versus standard media (Fig 1C) confirming proinflammatory effects of 9-(S)-HODE.

Previous research has shown that HODE in mammals can be generated from LA by free radical–triggered oxidation but also enzymatically due to the action of MPO, COX, and LOX enzymes (Kaduce et al, 1989; Zhang et al, 2002; Kubala et al, 2010). We, thus, used BLAST to identify the corresponding *Drosophila* enzymes with the goal of knocking-out function to reduce 9-HODE production. BLAST analysis demonstrated that *Drosophila* contains only MPO-like enzymes. No *Drosophila* LOX homologues were identified and, although a *Drosophila* COX–like enzyme (Pxt) has previously been reported (Tootle & Spradling, 2008), sequence comparison showed that Pxt is in fact more closely related to human MPO (BLAST E-values of 5.00E-77 for MPO versus 1.00E-10 for COX-1). This was confirmed by in vitro enzyme assay showing that purified FLAG-tagged Pxt possessed robust MPO activity that was abolished by the specific inhibitor 4-aminobenzoic hydrazide (MPO inhibitor-1) (Fig 2F).

We next examined whether reduction of MPO activity could influence the severity of inflammatory responses in flies of the

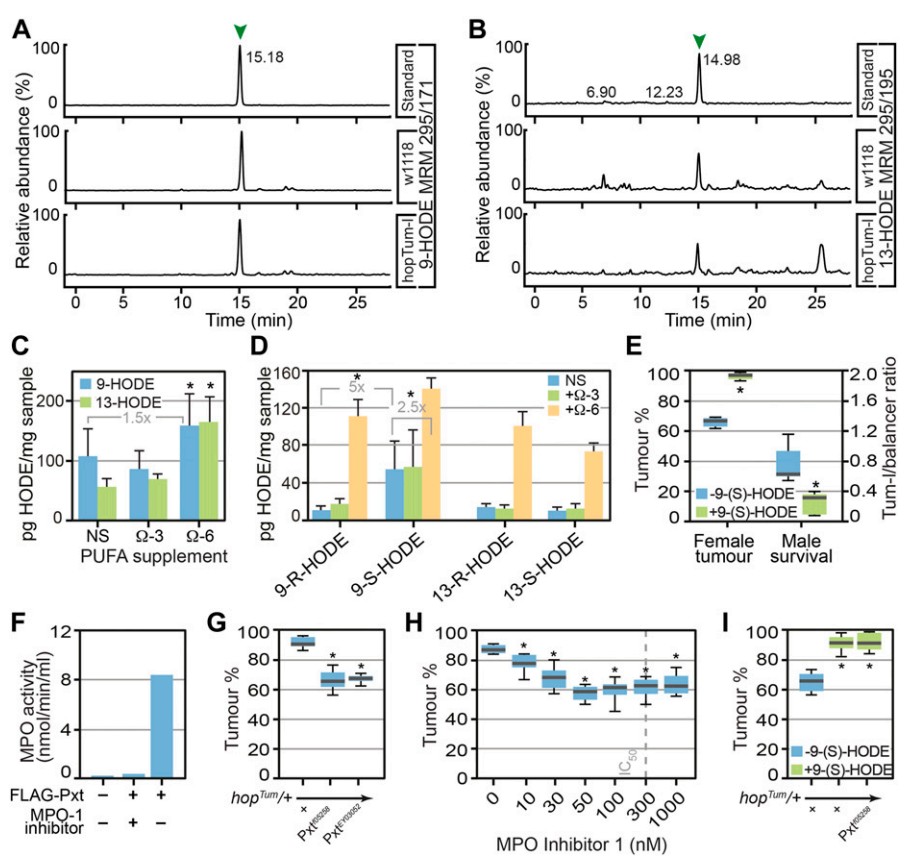

**Figure 2. *Drosophila* produce the lipid mediator 9-HODE which is required for inflammatory responses.**
Omega-6 PUFA LA can be processed either by COX, lipoxygenase, or myeloperoxidase (MPO) enzymes to produce hydroxy-octadecadienoic acids (HODEs). **(A, B)** 9-HODE (MRM m/z 295>171) and (B) 13-HODE (MRM m/z 295>195) were identified in *Drosophila* lipid extracts. **(C)** The concentrations of 9-HODE and 13-HODE were unaltered after ALA (omega-3 PUFA) supplementation but increased after LA supplementation (omega-6 PUFA). Data are mean and SD of three independent determinations. **(D)** Chiral LC-MS/MS shows that 9-(S)-HODE was the predominant species on non-supplemented (NS) and ALA supplemented medium, whereas LA supplementation promoted the production of both 9- and 13-HODE, at (R) and (S) configuration. **(E)** 9-(S)-HODE (1 $\mu$M) supplementation increased melanotic tumour incidence in $hop^{Tum}$ heterozygous females and increased lethality of $hop^{Tum}$ heterozygous males. **(F)** Purified Pxt exhibited MPO activity in vitro. 3xFLAG-tagged Pxt haem peroxidase domain was over-expressed in HeLa cells, purified, and assayed for MPO activity. MPO Inhibitor-1 was used to inhibit peroxidase activity. **(G)** Reduction in levels of Pxt using $Pxt^{f05258}$ and $Pxt^{EY03052}$ suppressed melanotic tumours. **(H)** MPO inhibitors suppressed melanotic tumours, IC$_{50}$ of MPO inhibitor-1 is indicated. For (G, H), tumour incidence was assayed in heterozygous COX/MPO mutant/$hop^{Tum}$ females raised at 27°C on medium supplemented with 0.15% LA. **(I)** Increased melanotic tumour incidence in $hop^{Tum}$ heterozygous females raised on 9(S)-HODE-containing media (1 $\mu$M) was not suppressed by reduction in Pxt levels using $Pxt^{f05258}$. Data information: in (C, D), three replicates were used for each assay point. * indicates values statistically significantly different from control, $P$-value < 0.05 determined using $t$ test. In (E, F, G, H, I) 10 replicate crosses were used for each assay point. Box and whiskers plots were generated using R. * indicates values statistically significantly different from unsupplemented, $P$-value < 0.001 determined using $t$ test. NS indicates not statistically significantly different.

$hop^{Tum}$ inflammation model raised on LA-supplemented media. Genetic reduction in MPO levels using the transposon mutant alleles $Pxt^{f05258}$ and $Pxt^{EY03052}$ decreased tumour incidence (Fig 2G). This was replicated by pharmacological blockade of MPO activity by MPO inhibitor-1 treatment, which caused equivalent decreases in tumour incidence (Fig 2H). However, when media was supplemented with 9-(S)-HODE (the product of MPO oxidation of LA), genetic reduction in MPO levels using $Pxt^{f05258}$ mutants failed to decrease tumour incidence (Fig 2I). Taken together, these results were consistent with MPO-mediated oxidation of the omega-6 PUFA LA generating 9-(S)-HODE that in turn modulates inflammatory responses.

To test whether HODEs, such as other oxylipins, have inflammatory functions to combat infection and mediate tissue repair, we examined whether mutation of $Pxt$ affected survival in response to wounding and infection. Kaplan–Meier survival curves demonstrated marked decrease in the lifespan of $Pxt$ mutants, with a mutant lifespan of 60% of that of controls (Fig S2A). This was reflected by reduced survival of $Pxt$ mutants in response to sterile inflammation (Fig S2B) and fungal infection (Fig S2C). The latter, in particular, is consistent with a large body of evidence indicating MPO deficiency in humans is associated with enhanced susceptibility to fungal infections (Lehrer & Cline, 1969).

### Identifying 9-(S)-HODE targets by transcriptional profiling

To clarify further the mechanism by which 9-(S)-HODE acts, we used transcriptional profiling to identify downstream gene targets, and thus identify regulatory networks controlled by 9-(S)-HODE. *Drosophila* S2 cells were treated with 9-(S)-HODE or vehicle alone for 12 and 24 h and set of 325 and 224 genes that were respectively up- or down-regulated at both time-points was identified by microarray analysis (Fig 3A). Although our initial prediction was that 9-(S)-HODE treatment would affect genes with known proinflammatory signaling

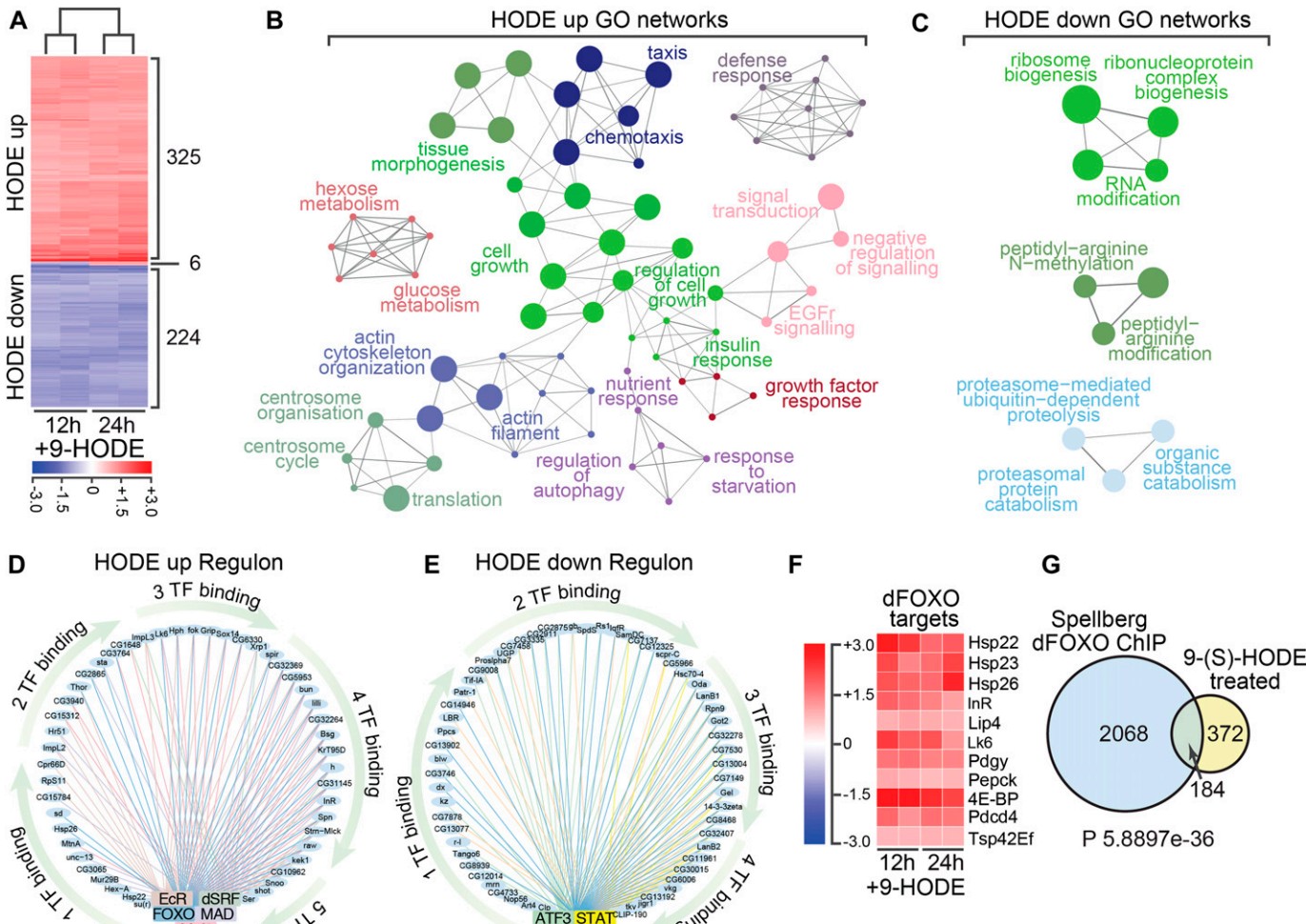

**Figure 3. Transcriptome analysis reveals 9-(S)-HODE regulates dFOXO targets.**
**(A)** Heat map showing differentially expressed genes in S2 cells 12 and 24 h after 9-(S)-HODE treatment. Scale bar indicates fold-change relative to mock-treated control.
**(B)** Gene ontology (GO) network analysis of the 325 genes significantly up-regulated in S2 cells 24 h after 9-(S)-HODE treatment. **(C)** GO network analysis of the 224 genes significantly down-regulated in S2 cells 24 h after 9-(S)-HODE treatment. **(D, E)** iRegulon analysis of up- and down-regulated genes defines TF regulatory circuits that transduce 9-(S)-HODE signals. Binding sites for dFOXO are a common feature of both sample sets. **(F)** Heat map showing fold-change in transcript levels of known dFOXO target genes after 9-(S)-HODE treatment. **(G)** Venn diagram showing overlap between 9-(S)-HODE transcriptional targets and genes previously defined as dFOXO targets by chromatin immunoprecipitation (ChIP) on chip analysis (Spellberg dFOXO ChIP). Hypergeometric test was performed in R to determine significance of overlap.

functions, most targets were in fact related to control of growth, metabolism, and proliferation (Fig S3A). Pathway–target networks were constructed from these differentially expressed genes using known *Drosophila* protein–protein and genetic interactions (BioGRID) and defined key target processes as insulin signaling, ribosome biogenesis, and translation control, confirming activities in growth control (Fig S3B). This was confirmed by using the Cytoscape plug-in ClueGO (Bindea et al, 2009) and GO biological process annotations to cluster all differentially regulated genes, as well as separately up- or down-regulated transcripts into regulatory networks. Pathways identified to be up-regulated after 9-(S)-HODE-treatment showed functions in insulin signaling, control of growth, tissue morphogenesis, and hexose metabolism (Fig 3B), whereas down-regulated functional clusters predominantly corresponding to proteasome function, ribosome organisation, and protein methylation (Fig 3C).

### 9-HODE targets overlap the *Drosophila* FOXO (dFOXO) regulome

To define the regulatory hierarchy that underpins the 9-(S)-HODE-transcriptional response, we used the iRegulon computational framework (Janky et al, 2014) to analyse the cis-regulatory elements of differentially regulated genes to identify enriched motifs for known TFs in the two classes of targets. Parallel analysis of 9-(S)-HODE-up (Fig 3D) and 9-(S)-HODE-down (Fig 3E) gene targets identified a restricted set of TFs that control transcription of 9-(S)-HODE-targets. These sets were non-overlapping, with one exception: binding sites for the dFOXO TF were detected in both up- and down-regulated gene targets. Additional TFs identified include MINICHROMOSOME MAINTENANCE1, AGAMOUS, DEFICIENS and SERUM RESPONSE FACTOR (MADS), Drosophila serum response factor, Mef2, and EcR for up-regulated targets (Fig 3D) and ATF3, STAT, and GATA for down-regulated targets (Fig 3E).

Consistent with dFOXO involvement in the 9-(S)-HODE-transcription response, analysis of microarray expression of previously defined direct dFOXO targets (Puig et al, 2003; Teleman et al, 2008; Vihervaara & Puig, 2008; Olson et al, 2013; Donovan & Marr, 2016) revealed that all were up-regulated after 9-(S)-HODE-treatment (Fig 3F). Previous studies have defined ~2,200 dFOXO targets in S2 cells by ChIP (chromatin immunoprecipitation) on chip analysis of cells that express a constitutively active dFOXO variant (Spellberg & Marr, 2015). In principle, this allows the full-spectrum of dFOXO targets to be discriminated. By comparing our 9-(S)-HODE differentially regulated genesets with putative dFOXO targets defined by ChIP on chip (Spellberg & Marr, 2015), we show that 33% of our 9-(S)-HODE targets were dFOXO ChIP targets (Fig 3G), suggesting that they are directly regulated by dFOXO.

### 9-HODE activates dFOXO

Enrichment of dFOXO targets in 9-(S)-HODE differentially regulated genes suggests that dFOXO underpins the 9-(S)-HODE-transcriptional response. dFOXO is the nexus at which the IIS pathway and cellular stress response pathways interface to control growth and metabolism. In normal growth, IIS activates the kinase Akt. Akt phosphorylation down-regulates dFOXO by triggering nuclear export and/or proteolysis. Conversely, under conditions of stress, kinases such as c-Jun N-terminal kinase (JNK) phosphorylate distinct sites on dFOXO causing nuclear accumulation and stabilisation that overrides the effect of Akt

(Essers et al, 2004; Wang et al, 2005; Heimbucher et al, 2015) and initiating a stress-specific transcriptional programme.

By controlling dFOXO activation, 9-(S)-HODE would be expected to have widespread effects not only on stress responses but also on metabolism and growth. It has previously been shown that expression of a constitutively activated dFOXO variant in S2 cells induces growth arrest (Puig et al, 2003). We, therefore, tested whether 9-(S)-HODE treatment affected growth and proliferation of S2 cells. S2 cells were seeded into fresh medium containing either 9-(S)-HODE or vehicle alone and cultured for 3 d. Consistent with 9-(S)-HODE activating dFOXO and inhibiting proliferation, cell number was reduced after 9-(S)-HODE treatment at both 1 and 3 d after treatment (Fig 4A). This was accompanied by cell cycle arrest as evidenced by FACS analysis of propidium iodide (PI)–stained control and 9-(S)-HODE–treated S2 cells (Fig 4B). Consistent with 9-(S)-HODE stabilising dFOXO, Western analysis showed elevated dFOXO levels after 9-(S)-HODE treatment (Fig 4C). This was confirmed by confocal immunofluorescence of S2 cells showing high levels of nuclear dFOXO after treatment (Fig 4D and E).

In an effort to discriminate the kinetics of this response, we used a dFOXO mCherry knock-in (*dFOXO-mCherry*) (Kakanj et al, 2016), to monitor dFOXO distribution in live tissue. As shown in Fig 4F, 9-(S)-HODE treatment triggered nuclear accumulation of dFOXO-mCherry in the *Drosophila* fat body (the fly equivalent of adipose tissue and liver). Live imaging revealed rapid nuclear accumulation of within 10 min of 9-(S)-HODE treatment (Fig 4F). In contrast, the related LA oxidation product 13-(S)-HODE did not cause nuclear dFOXO-mCherry accumulation (Fig S4A and B). dFOXO nuclear accumulation triggered by 9-(S)-HODE was accompanied by recruitment of dFOXO to known DNA targets. ChIP revealed dFOXO binding to previously defined dFOXO targets in 9-(S)-HODE–treated S2 cells (Fig 4G). Thus, binding to the promoters of small heat shock genes as well as the promoter of Pdcd4 and the *Drosophila* 4E-BP, *Thor*, was increased after 9-(S)-HODE treatment (Fig 4G). No relative enrichment compared with untreated cells was detected at control genomic regions.

Analysis of the related human FOXO3 confirmed that 9-(S)-HODE–mediated FOXO activation was a conserved process. Thus, whereas transiently transfected HA-tagged FOXO3 was predominantly cytoplasmic, 9-(S)-HODE treatment triggered nuclear localisation of FOXO3-HA in HeLa cells (Fig 4H). FOXO3 activation by 9-(S)-HODE was confirmed by analysing expression of a luciferase reporter driven by three copies of a FOXO-binding site (FHRE-luc) (Brunet et al, 1999). As shown in Fig 4I, strong induction of luciferase activity was observed when cells expressing FOXO3 were also treated with 9-(S)-HODE for 24 h indicating that 9-(S)-HODE is able to stimulate FOXO transcriptional activity in human cells. By driving constitutively elevated FOXO nuclear levels, 9-(S)-HODE may override IIS inputs providing a mechanism by which insulin resistance could be established. To verify this, we tested whether dFOXO nuclear accumulation triggered by 9-(S)-HODE was observed after treatment with insulin. In S2 cells, 9-(S)-HODE triggered dFOXO nuclear localisation even in the presence of insulin (Fig 4J and K).

### Mechanism of dFOXO activation by 9-HODE

We next clarified the mechanism by which 9-(S)-HODE induces dFOXO nuclear accumulation. As stress signals acting via the JNK

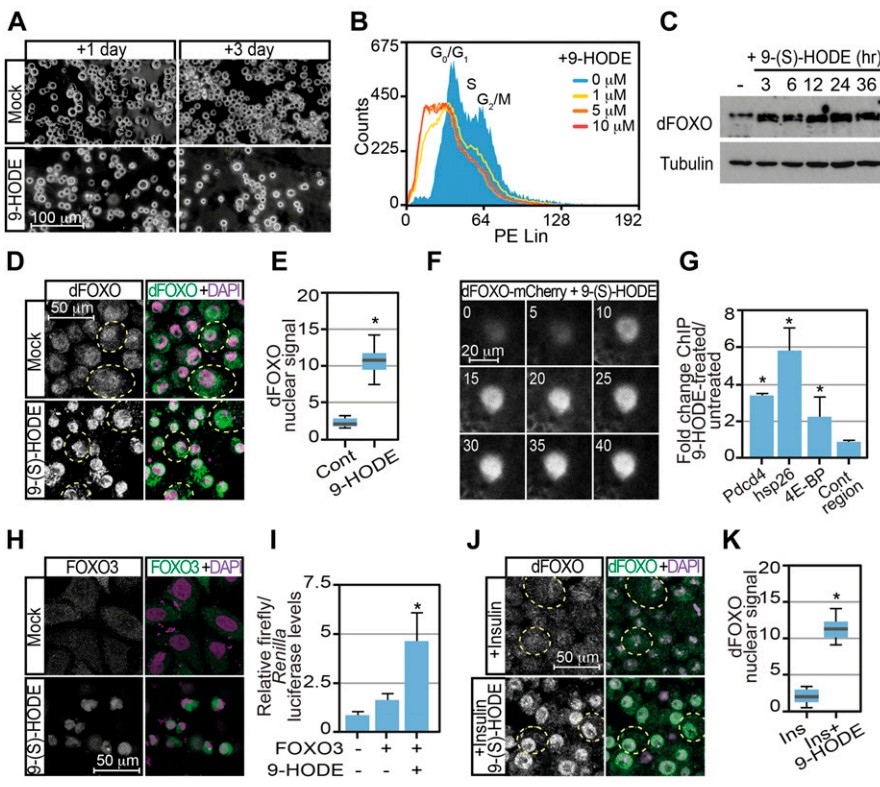

**Figure 4. 9-(S)-HODE activates dFOXO.**
**(A)** 9-(S)-HODE (10 μM) treatment inhibits *Drosophila* S2 cell proliferation. **(B)** FACS cell cycle analysis confirms cell cycle arrest. G1 and G2/M peaks seen in mock-treated S2 cells were absent in 9-(S)-HODE–treated cells replaced by a broad sub-G1 peak. **(C)** Western analysis of dFOXO levels in S2 cells 24 h after 9-(S)-HODE treatment. MAb E7 anti-tubulin was used as a loading control. **(D)** Immunofluorescence microscopy of *Drosophila* S2 cells shows elevated levels of nuclear dFOXO after 9-(S)-HODE treatment. **(E)** Quantitation of nuclear dFOXO signals in S2 cells after mock or 9-(S) HODE treatment. **(F)** Live imaging of 9-(S)-HODE–treated dFOXO-mCherry fat body confirming 9-(S)-HODE elevates nuclear dFOXO within 10 min of treatment. **(G)** Ratio of dFOXO ChIP signal at known dFOXO targets and control genomic regions in S2 cells treated with 9-(S)-HODE for 24 h relative to mock/untreated dFOXO ChIP signal. **(H)** Immunofluorescence microscopy of *Drosophila* S2 cells reveals elevated levels of nuclear dFOXO after 9-(S)-HODE treatment is resistant to insulin treatment. **(I)** Quantitation of nuclear dFOXO signals in mock or 9-(S) HODE treated S2 cells that are subsequently treated with insulin for 90 min before fixation. **(J)** 9-(S)-HODE treatment of human HeLa cells triggers nuclear entry of transiently transfected FOXO3-HA. **(K)** 9-(S)-HODE treatment of FOXO3-HA transfected human HeLa cells activates transcription from the FOXO3a reporter plasmid FHRE-luc. Data information: in all panels, * indicates values statistically significantly difference *P*-value < 0.001 determined using *t* test. NS, no significant difference. In (E, K), 50 determinations were used for each assay point. Box and whiskers plots were generated using R. In (G, I), three replicates were used for each assay point.

pathway can regulate FOXO nuclear levels, we tested whether JNK signaling was required for 9-(S)-HODE–stimulated responses, first examining effects on melanotic tumour production. As a control, we observed that reduction in dFOXO levels (by crossing one copy of the *foxo^Δ94* mutation into the *hop^Tum* background) significantly reduced melanotic tumour incidence (Fig 5A). Decreasing JNK signaling by deletion of the *Drosophila* JNK, *basket* (*bsk*), similarly reduced melanotic tumour incidence (Fig 5A). In contrast, increased JNK signaling by reduction in levels of the JNK inhibitor *puckered* (*puc*) enhanced melanotic tumour incidence (Fig 5A).

Further evidence that JNK transduces 9-(S)-HODE signals was provided by elevated levels of the phosphorylated (active) form of JNK (pJNK) in 9-(S)-HODE–treated cells (Fig 5B–E). This occurred in both *Drosophila* (S2) and human (HeLa) cells confirming that this process is conserved. Moreover, this JNK activation was required for 9-(S)-HODE–stimulated dFOXO nuclear accumulation as genetic ablation or chemical inhibition of JNK blocked dFOXO nuclear entry. Thus, over-expression of *bsk* RNAi or dominant negative Bsk variants (Bsk-DN) using the fat body–specific *FB-GAL4* driver was able to reduce 9-(S)-HODE–stimulated dFOXO-mCherry nuclear entry in third instar fat body tissue (Fig 5F and G). Similarly, treatment with the JNK inhibitor SP600125 was able to block 9-(S)-HODE–triggered nuclear entry of dFOXO in S2 cells (Fig 5H and I). Importantly, neither *bsk* RNAi nor Bsk-DN expression in the fat body reduced dFOXO levels as cytoplasmic dFOXO-mCherry signal was indistinguishable from otherwise wild-type backgrounds in the absence of 9-HODE (Fig S4C). However, cytoplasmic dFOXO-mCherry levels were reduced in wild-type fat body after treatment with 9-(S)-HODE,

accompanied by concomitant increases in nuclear dFOXO-mCherry levels (Fig S4D).

Finally, as additional controls, we also tested whether 9-(S)-HODE generates or requires reactive oxygen species (ROS) to induce dFOXO nuclear localisation of S2 cells. CellROX green staining revealed no increase in ROS in 9-(S)-HODE–treated S2 cells (Fig 5J). Moreover, ROS was not required for 9-(S)-HODE–triggered dFOXO nuclear localisation as treatment with the free radical scavenger N-acetyl-L-cysteine (NAC) failed to prevent nuclear entry after 9-(S)-HODE treatment (Fig 5K). Taken together with our previous results, these data suggest a trajectory in which oxidation of the omega-6 PUFA LA generates 9-(S)-HODE, in turn activating JNK that stabilises nuclear dFOXO to control expression of FOXO targets.

## Discussion

Obesity is closely linked to a suite of conditions, including type 2 diabetes and cardiovascular disease that are grouped under the broad moniker of "metabolic syndrome." Core underlying components of metabolic syndrome are low-grade metabolically triggered inflammation (meta-inflammation) coupled with excessive flux of fatty acids. Here, we use a *Drosophila* inflammatory model to delineate how changes in fatty acid balance and levels impact meta-inflammation and in turn modulate insulin signaling pathways. We demonstrate that the dietary omega-6 PUFA LA is metabolised to produce 9-(S)-HODE which, in turn, enhances inflammatory responses.

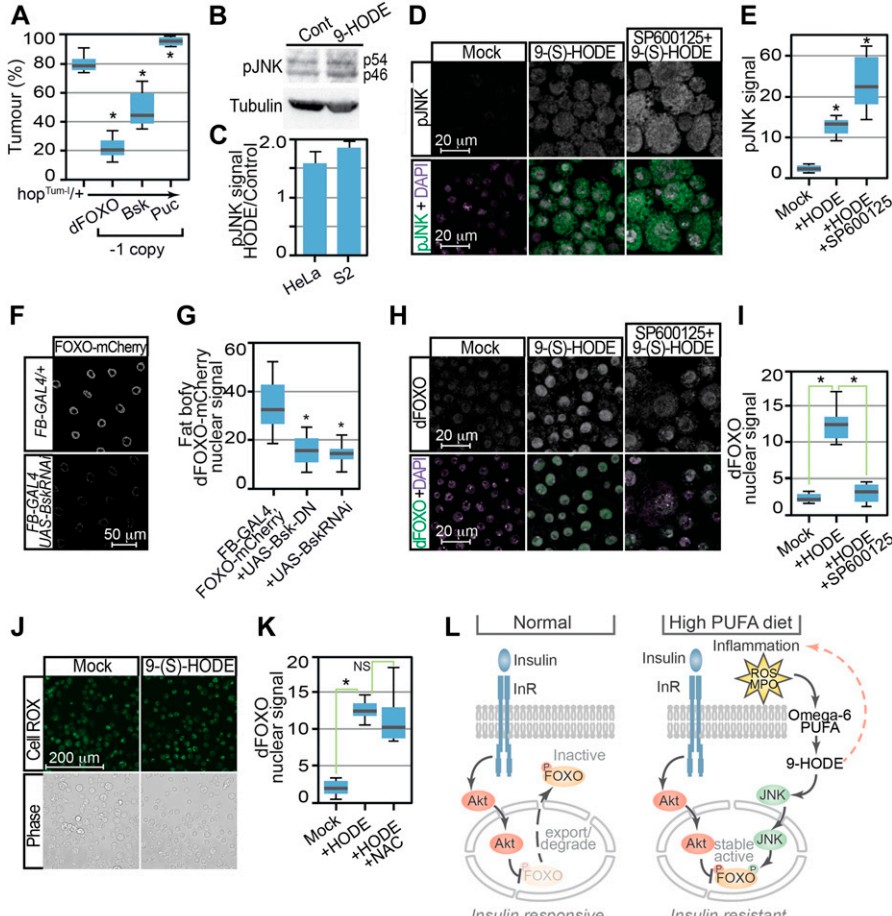

**Figure 5. Control of dFOXO by 9-HODE is mediated via the JNK pathway.**
**(A)** Tumour incidence in heterozygous *hop^Tum* females flies was decreased by reduction in dFOXO (*Foxo^Δ94*) or Bsk (*bsk^1*) levels but increased by removal of Puc (*puc^E69*). **(B)** JNK activation was confirmed by Western analysis of human HeLa and *Drosophila* S2 cells 24 h after 9-(S)-HODE treatment using anti-pJNK antibodies. MAb E7 anti-tubulin is used as a loading control. Expected pJNK (p54 and p56) species in S2 extracts are indicated. **(C)** Quantitation of pJNK levels in 9-(S)-HODE–treated cells relative to mock-treated control. **(D)** Confocal immunofluorescence microscopy of mock and 9-(S)-HODE–treated S2 cells using anti-pJNK anti-dFOXO antibodies confirms JNK activation after 9-(S)-HODE treatment. **(E)** Quantitation of pJNK signals in S2 cells. **(F)** Fat body–specific over-expression of dominant-negative Bsk variants or RNAi-mediated inhibition *bsk* reduces 9-(S)-HODE stimulated dFOXO-mCherry nuclear entry. **(G)** Quantitation of dFOXO-mCherry nuclear signal in fat body tissue. **(H)** Pretreatment with JNK inhibitor SP600125 prevents nuclear dFOXO localisation after 9-(S)-HODE treatment. **(I)** Quantitation of dFOXO nuclear signal in S2 cells after treatment with 9-(S) HODE alone or with the JNK inhibitor SP600125. **(J)** 9-(S) HODE treatment of S2 cells does not increase reactive oxygen species as revealed by CellROX staining. **(K)** Reactive oxygen species are not required for 9-(S)-HODE–triggered dFOXO nuclear localisation as simultaneous treatment with N-acetyl-L-cysteine (NAC) does not prevent nuclear entry. **(L)** Proposed model indicating how 9-HODE activation of FOXO could confer insulin resistance. Data information: in (A), 10 replicate crosses were used for each assay point. Box and whiskers plots were generated using R. * indicates values statistically significantly different from unsupplemented, *P*-value < 0.001. In (E, G, I, K), 50 determinations were used for each assay point. Box and whiskers plots were generated using R. * indicates values statistically significantly different from unsupplemented, *P*-value < 0.001 determined using *t* test.

Moreover, in addition to being a modulator of inflammation, 9-(S)-HODE also activates dFOXO via the JNK pathway. By activating dFOXO, a key transducer of the IIS pathway, 9-HODE provides a novel link between fatty acid balance, inflammation, and control of insulin signaling.

Our dietary supplementation experiments reveal clear pro- and anti-inflammatory effects mediated by the omega-6 and omega-3 PUFAs LA and ALA, respectively. These data agree with numerous mammalian studies showing production of pro- and anti-inflammatory lipid mediators from omega-6 and omega-3 PUFAs, respectively, as well as studies linking increased incidence of inflammatory disease to changes in the balance of omega-6 and omega-3 PUFAs in Western diets (Simopoulos, 2002). Although prostaglandins are important proinflammatory mediators of omega-6 PUFAs in mammals, we were unable to detect prostaglandins or their precursor AA (C20 PUFA) in *Drosophila* larvae. This may reflect an inability of *Drosophila* to elongate PUFA carbon chains beyond C18, agreeing well with previous fatty acid analyses of *Drosophila* (Yoshioka et al, 1985; Teague et al, 1986; Shen et al, 2010) and suggests that *Drosophila* lack endogenously generated eicosanoid lipid mediators.

Instead, we show that in *Drosophila* the main dietary omega-6 PUFA, LA, is metabolised to HODE. We identified 9-(S)-HODE as the predominant LA metabolite, showed that levels correlate with

increased severity of an inflammation model, and that treatment with 9-(S)-HODE enhances melanotic tumours, suggesting that HODEs mediate the proinflammatory effects of the omega-6 PUFA LA. We speculate that the oxidation of LA to generate 9-HODE is an ancient, conserved response to damage and infection. Although most mammalian studies focus on the role of lipid mediators such as the eicosanoids, we suggest that some of the proinflammatory effects of omega-6 PUFAs on humans may also be due to the production of 9-HODE. Certainly, increased levels of 9-HODE are associated with chronic inflammation (Jira et al, 1998; Waddington et al, 2001) and altered expression of proinflammatory markers (Mikita et al, 2001; Hattori et al, 2008). As they lack prostaglandins, *Drosophila* provide a simplified system in which HODE functions can be interrogated without the complication of other families of lipid mediators and offers a powerful experimental model to characterize HODE function and effectors.

A key target of 9-(S)-HODE identified by our research is dFOXO. 9-(S)-HODE acting via the JNK pathway triggers increased nuclear levels and chromatin binding of dFOXO. dFOXO is an important transducer of the IIS pathway indicating that 9-(S)-HODE, in addition to proinflammatory functions, has the potential to influence insulin signaling. Based on these results, we propose the following model (summarized in Fig 5L) to explain how fatty acid balance controls inflammation and also insulin signaling. Under conditions

where omega-6 and omega-3 PUFA levels are controlled and in balance, IIS functions normally to down-regulate dFOXO by triggering nuclear export and/or degradation. In contrast, excess consumption of omega-6 PUFA, in the presence of inflammation, induces lipid peroxidation and the release of 9-(S)-HODE. In turn, 9-(S)-HODE enhances inflammation, further stimulating lipid peroxidation and 9-(S)-HODE production. Elevated 9-(S)-HODE also activates dFOXO via the JNK pathway. By driving constitutively nuclear dFOXO, elevated levels of 9-(S)-HODE have the potential to override IIS inputs, providing a molecular mechanism by which insulin resistance could be established. Data presented here provide some support for this model as we show using cultured cells that 9-(S)-HODE pretreatment prevents nuclear export/degradation of dFOXO in the presence of insulin. However, further analysis of the effects of 9-(S)-HODE on the metabolism and growth of flies are required to demonstrate conclusively that all physiological cognates of insulin resistance occur after of 9-(S)-HODE treatment.

Nevertheless, by showing that oxidation of LA generates 9-(S)-HODE, in turn activating JNK and FOXO, our results identify a key mediator that potentially links changes in fatty acid balance and inflammation to insulin resistance. As such, regulation of the production and activity of 9-(S)-HODE offers a convenient avenue for therapeutic intervention to prevent the development of insulin resistance. Data presented here suggest that one approach may be the control of dietary balance of omega-6 and omega-3 PUFAs. We observed that the omega-3 PUFA ALA overrides the inflammatory effect of LA (omega-6 PUFA) even when present at sub-stochiometric levels. ALA can be metabolised by the same enzymes and pathways as omega-6 LA. However, instead of producing hydroxy-octadecadienoic acids (HODEs), oxidation of ALA generates hydroxy-octadecatrienoic acids (HOTrE), 9-, and 13-HOTrE. HOTrEs have been detected in humans (Shearer et al, 2010); however, little is known about their function. We show here that ALA has anti-inflammatory actions, suppressing melanotic tumours in the $hop^{Tum}$ inflammation model. Our data make it a priority to determine HOTrE prevalence and targets, in particular the relative effects of HODEs and HOTrEs on JNK and FOXO activation, and the receptors for these compounds.

More broadly, our data integrate well with the emerging consensus that controlling lipid oxidation has a widespread protective function in the cellular response to stress and metabolic imbalance. In particular, shielding PUFAs against oxidative damage through lipid droplet formation is an integral response to damage-related and metabolic disease states (Walther & Farese, 2012). Lipid droplets accumulate in *Drosophila* larval brains under conditions of oxidative stress (Bailey et al, 2015). Neurodegeneration occurs when this protection is overwhelmed (Liu et al, 2015). Human macrophages, especially those in atherosclerotic plaques, also accumulate lipid droplets, and increasing their capacity to store triglycerides disconnects obesity and diabetes (Walther & Farese, 2012). It is, thus, noteworthy that a hallmark of our $hop^{Tum}$ inflammation model is the accumulation of lipid droplets in macrophages (MW and PB, unpublished data). Although these may initially serve to protect LA against oxidation, we speculate that concerted inflammation overwhelms this capacity, triggering the release of 9-(S)-HODE, leading to subsequent JNK and FOXO

activation. Cumulatively, these data define a disease trajectory that potentially links imbalances in omega-6 PUFA levels and elevated inflammation to damage and metabolic disease through the release of HODE lipid mediators.

# Materials and Methods

### *Drosophila* strains

The $hop^{Tum}$ strain is as described by Hanratty and Dearolf (1993). An isogenic $w^{1118}$ strain (Badenhorst et al, 2002) was used as a control. $Pxt^{f05258}$ and $Pxt^{EY03052}$ were obtained from the Bloomington Drosophila Stock Center. $dFOXO^{94}$ (Slack et al, 2011) and *dFOXO-mCherry*, an mCherry knock-in the endogenous *dFOXO* locus (Kakanj et al, 2016) were provided by Linda Partridge. $bsk^1$ and $puc^{E69}$, *UAS-bsk-DN*, and $UAS-bsk-RNAi^{GD10555}$ were provided by Yun Fan. *FB-GAL4* (Gronke et al, 2003) was provided by Ines Anderl.

### Cell lines

HeLa cells (CCL2 isolate, RRID: CVCL_0030) were obtained from the ATTC and STR profile verified (Amelogenin: X, CSF1PO: 9,10; D13S317: 12,13.3; D16S539: 9,10; D5S818: 11,12; D7S820: 8,12; THO1: 7; TPOX: 8,12; vWA: 16,18). HeLa Cells were incubated at 37°C with 5% $CO_2$ and cultured in DMEM containing 10% FBS, 100 U penicillin–streptomycin (Sigma-Aldrich), 1 mM glutamine (GIBCO), and 1× nonessential amino acids (Sigma-Aldrich). *Drosophila* Schneider 2 (S2) cells were obtained from Invitrogen (RRID: CVCL_Z232). Cell stocks have been tested for contamination of bacteria, yeast, mycoplasma, and virus and characterized by isozyme and karyotype analysis. S2 cells were cultured at 25°C in protein-free medium containing L-glutamine (Insect-express; Lonza), supplemented with 10% FBS, 100 U penicillin–streptomycin (Sigma-Aldrich).

### *Drosophila* inflammatory tumour assay

*Drosophila* were raised on dextrose, wheat flour, yeast, and agar media. For melanotic tumour assays, five $hop^{Tum}$/FM7 females (Luo et al, 1995) were crossed with either five $w^{1118}$ males as a control or five mutant males (in the case of mutant enhancer/suppressor studies) per vial. Crosses were allowed to mate for 2 d before being transferred to fresh media. Five initial vials were set up for each datum point and were incubated for 3 d before being transferred to a second fresh set of vials, giving a total of 10 replicates per experiment. The total number of animals assayed for each data-point was in the range 250–300. Required sample size was estimated using a statistical power calculator (https://www.stat.ubc.ca/~rollin/stats/ssize/n2.html) assuming an error rate of 0.01 and required power 1.00. Crosses were incubated for 3 d and the adult flies then removed. $hop^{Tum}$ females of the next generation were scored for tumour incidence using a Nikon SMZ645 stereomicroscope. All crosses were maintained on standard media or supplemented media where stated, and all crosses were incubated at 27°C unless otherwise stated.

# Key Resources Table.

| Reagent or resource | Source | Identifier |
|---|---|---|
| Antibodies | | |
| Mouse anti-B-tubulin MAb E7 | Developmental Studies Hybridoma Bank | RRID:AB_528499 |
| Mouse anti-phospho-SAPK/JNK (Thr183/Tyr185) MAb G9 | Cell Signalling Technology | RRID:AB_2307321 |
| Rabbit anti-dFOXO | Michael Marr | BRD10 |
| Cy3-conjugated AffiniPure goat anti-rabbit antibody | Jackson ImmunoResearch | Cat. no. 111-167-003 |
| FITC-conjugated AffiniPure donkey anti-mouse antibody | Jackson ImmunoResearch | Cat. no. 715-095-151 |
| Anti-rabbit HRP-conjugated IgG (H+L) | GE Healthcare | |
| Anti-mouse HRP-conjugated IgG (H+L) | GE Healthcare | |
| Rat monoclonal anti-HA-tag [7C9] | ChromoTek | |
| Deposited data | | |
| Agilent microarray data | This article | Array express: E-MTAB-6253 |
| Experimental models: organisms/strains | | |
| *Drosophila melanogaster: Schneider 2, CVCL_Z232* | Invitrogen | RRID: CVCL_Z232 |
| *Homo sapiens: HeLa CCL-2* | American Type Culture Collection (ATCC) | RRID: CVCL_0030 |
| *D. melanogaster: hop$^{Tum}$* | Bloomington Drosophila Stock Center | RRID: BDSC_8492 |
| *D. melanogaster: w$^{1118}$* | Bloomington Drosophila Stock Center | RRID: BDSC_5905 |
| *D. melanogaster: w$^{1118}$; foxo$^{Δ94}$/TM6B, Tb$^1$* | Bloomington Drosophila Stock Center | RRID: BDSC_42220 |
| *D. melanogaster: w$^{1118}$ P{w+mC=UAS-bsk.DN}2* | Bloomington Drosophila Stock Center | RRID: BDSC_6409 |
| *D. melanogaster: bsk$^1$ cn$^1$ bw$^1$ sp$^1$/CyO,* | Bloomington Drosophila Stock Center | RRID: BDSC_3088 |
| *D. melanogaster: w\*; cno$^3$ P{ry+t7.2=A92}pucE69/TM6B, P{w+mC=iab-2(1.7)lacZ}6B, Tb$^1$* | Bloomington Drosophila Stock Center | RRID: DGGR_109029 |
| *D. melanogaster: w$^{1118}$; P{GD10555}v3413 UAS-bskRNAi* | Bloomington Drosophila Stock Center | RRID:FlyBase_FBst0460476 |
| *D. melanogaster: w$^{1118}$; P{GAL4}fat (FB-GAL4)* | Gronke et al (2003) | RRID:FlyBase_FBti0013267 |
| *D. melanogaster: w$^{1118}$; P{GD10555}v3413 UAS-bskRNAi* | Bloomington Drosophila Stock Center | RRID:FlyBase_FBst0460476 |
| *D. melanogaster: y$^1$ w$^{67c23}$; P{EPgy2}Pxt$^{EY03052}$* | Bloomington Drosophila Stock Center | RRID:BDSC_15620 |
| *D. melanogaster: PBac{WH}Pxt$^{f05258}$* | The Exelixis Collection at the Harvard Medical School | RRID: FBti0067872 |
| *D. melanogaster: dfoxo-v3-mCherry* | Kakanj et al (2016) | N/A |
| *Beauveria bassiana* | CABI | (T. Petch, 1947) IMI: 12942 |
| Oligonucleotides | | |
| foxo-ChIP-control forward primer (5'-AGCCCGCGAAGATACAAGAG-3') | This article | N/A |
| foxo-ChIP-control reverse primer (5'-AGCCACAAACAGCGACAGAA-3') | This article | N/A |
| *pdcd4*-ChIP forward primer (5'-CCGTTGGGAGTCTCTCTCTC-3') | This article | N/A |
| *pdcd4*-ChIP reverse primer (5'-GCTCTCCCCACCTTCTCACC-3') | This article | N/A |
| *hsp26*-ChIP forward primer (5'-GTGCGCCTGTATGAGTGAGA-3') | This article | N/A |
| *hsp26*-ChIP reverse primer (5'-GTGGGAGATTGCTGGCGTTA-3') | This article | N/A |
| *4EBP*-ChIP forward primer (5'-CAAGAACCAGCCGGTTTGTC-3') | This article | N/A |

**Life Science Alliance**

**Continued**

| Reagent or resource | Source | Identifier |
|---|---|---|
| *4EBP*-ChIP reverse primer (5′-GCTCTCTTCTCGCTCTTT CG-3′) | This article | N/A |
| *Pxt*-RT-PCR forward primer (5′-GCAGCTCCTCGATGTGATTGAAAC-3′) | This article | N/A |
| *Pxt*-RT-PCR reverse primer (5′-CTAGAGTGCGAGCGAGAGGTAAGA-3′) | This article | N/A |
| *RpL32*-RT-PCR forward primer (5′-ATCCGCCACCAGTCGCATCGATATGCTAAG-3′) | This article | N/A |
| *RpL32*-RT-PCR reverse primer (5′-TCTTGAGAACGCAGGCGACCGTTGGGGTTG-3′) | This article | N/A |
| *DptA*-RT-PCR forward primer (5′-GCAGTTCACCATTGCCGTCGCCTTACTTTG-3′) | This article | N/A |
| *DptA*-RT-PCR reverse primer (5′-TGAAGATTGAGTGGGTACTGCGGTGGTGGA-3′) | This article | N/A |
| *Drs*-RT-PCR forward primer (5′-ATCAAGTACTTGTTCGCCCTCTTCGCTGTC-3′) | This article | N/A |
| *Drs*-RT-PCR reverse primer (5′-CTCGTTGTCCCAGACGGCACAGGGACCCTT-3′) | This article | N/A |
| Recombinant DNA | | |
| Plasmid: pTRE3G | Clontech | 631173 |
| pTRE3G-3XFLAG | This article | N/A |
| pTRE3G-3XFLAG-Pxn | This article | N/A |
| Software and algorithms | | |
| FIJI | Schindelin et al (2012) | https://imagej.net/Fiji |
| Marray | Paquet (2009) | https://bioconductor.org/packages/release/bioc/html/marray.html |
| Limma | Ritchie et al (2015) | http://bioconductor.org/packages/release/bioc/html/limma.html |
| Bioconductor version 3.4 | http://bioconductor.org | https://bioconductor.org/news/bioc_3_4_release/ |
| R version 3.1.2 | https://www.R-project.org | https://cran.r-project.org/bin/windows/base/old/3.1.2/ |
| ClueGo | (Bindea et al, 2009) | https://apps.cytoscape.org/apps/cluego |
| esyN | (Bean et al, 2014) | http://www.esyn.org |
| Cytoscape | (Shannon et al, 2003) | https://cytoscape.org |
| iRegulon | (Janky et al, 2014) | http://iregulon.aertslab.org |

### *Drosophila* larval fatty acid supplementation

The following oils where added to the standard *Drosophila* medium before pouring into vials. As omega-6 PUFA source: safflower oil (Clear spring) containing 79% LA, 11% oleic acid (C18:1cis [n-9]), and 7% palmitic acid (C16 SFA). As omega-3 PUFA source: flaxseed oil (Food Supplement Company) containing 55% of α-linolenic acid, 18% oleic acid and 17% LA. As SFA source: coconut oil (Sigma-Aldrich) containing 92% SFA (including 48% lauric [C12], 20% myristic [C14], and 9% palmitic [C16] acids), with 6% monounsaturated FA and 1.7% PUFA (LA). Tumour assays using *hop^Tum^*/*w^1118^* female progeny were conducted as described above at 25°C (to detect enhancers) and 27°C (to detect suppressors). As the *hop^Tum^* mutant is temperature-sensitive-conditional, by varying culture temperature, the strength of JAK activation and baseline tumour incidence can be modified allowing both enhancers and suppressors to be assayed. Crosses were raised in the constant presence of the respective FAs under investigation. 9-(S)-HODE was supplemented to 1 $\mu$M final concentration in media to assay effects on melanotic tumour formation.

### MPO inhibitor treatment

The inhibitor myeloperoxidase inhibitor 1 (Calbiochem) was added to *Drosophila* culture medium supplemented with 0.15% safflower oil at concentrations spanning the published IC$_{50}$ (Kettle et al, 1995). Inhibitor was dissolved in DMSO (Sigma-Aldrich) and then added to fly media before pouring. Melanotic tumour assays using *hop^Tum^*/*w^1118^* female progeny were conducted at 27°C as described above. Media supplemented with vehicle alone provided reference control.

## Microarray analysis

For 9-HODE treatment, confluent S2 cells were split 1:4 to each well of a six well plate followed by addition of either 10 $\mu$M 9-(S)-HODE (Cayman Chemical) or equivalent volume of ethanol (vehicle only control). The cells were cultured for either 12 or 24 h and mRNA extracted using a $\mu$MACS mRNA Isolation kit (Miltenyi Biotec). The samples were assessed using an Agilent Bioanalyser 2100 with Nano RNA chip, and mRNA was dye-labeled (Cy3 for Control and Cy5 for HODE-treated) using the Agilent Low Input Quick Amp labeling method. Unbound fluorophore was removed using QIAGEN RNeasy columns and pairs of labeled samples mixed and fragmented in the presence of Agilent blocking reagents at 60°C for 30 min. The samples were hybridised to Agilent *Drosophila* Gene Expression Microarray, 4×44K arrays at 65°C for 17 h at a speed of 10 rpm. Slides were washed using the Agilent supplementary wash procedure and slides scanned using an Agilent Scanner C at 5-$\mu$m resolution using Agilent Scan Control Software 8.5 and the AgilentHD_GX_2 Color profile. The image was then imported into Agilent's Feature Extraction 10.10 software and the feature extraction was performed.

Statistical analysis was performed using R version 3.1.2 (http://www.R-project.org) and the *marray* and *limma* libraries of Bioconductor version 3.4 (http://www.bioconductor.org). Differential expression of genes was determined using an empirical Bayes approach within *limma* (Ritchie et al, 2015), with factors Control, HODE-12, and HODE-24. Moderated *t* statistics based on shrinkage of the estimated sample variance toward a pooled estimate and the corresponding *P* values were calculated for the HODE-12 versus Control and HODE-24 versus Control comparisons. *P* values were adjusted according to Benjamini & Hochberg (1995) to control the false discovery rate, and a threshold of 0.05 was used to select probe sets. Probe sets with statistically significant changes relative to wild-type were determined using the *topTable* function in *limma*. Genes with altered expression at respectively HODE 12-h treatment and HODE 24-h treatment are listed in Supplemental Datas 1 and 2. Array datasets are available through ArrayExpress (accession number E-MTAB-6253). GO network analysis was performed using the ClueGO Cytoscape plugin (Bindea et al, 2009). Size of nodes reflects significance of enrichment. Gene network analysis was performed using the esyN network tool (Bean et al, 2014) and nodes coloured according to array expression data using Cytoscape (Shannon et al, 2003). Regulatory networks of HODE up- and down-regulated genes were identified using the iRegulon app (Janky et al, 2014) in Cytoscape (Shannon et al, 2003) and nodes displayed in degree sorted circle format.

## Cell cycle analysis

S2 cells were grown in six-well plates as described above and treated with vehicle alone or increasing concentrations of 9-(S)-HODE for 24 h. Cells were pelleted, resuspended in 1 ml 1× PBS, and fixed using 4% formaldehyde (Sigma-Aldrich) for 10 min. Cells were washed twice using 1× PBS, resuspended in 1 ml PI/RNase staining buffer (BD Pharmingen), and incubated at room temperature for 30 min before analysis on a Beckman Coulter Cyan ADP analyser. The S2 population was gated using forward scatter and side scatter and PI staining quantitated using 800 V setting. PI staining distribution was determined using Summit V4.3.02 analysis software.

## Lipid extraction

200 mg of larvae samples was thawed and transferred to a Dounce glass homogeniser. Samples were homogenised in 500 $\mu$l ice-cold methanol using a glass pestle and mortar. The homogenate was transferred to a glass vial. The mortar was washed twice with 100 $\mu$l methanol and added to the homogenate. Ice-cold water was added to make a final 15% vol/vol methanol:water homogenate. 20 $\mu$l of 1 ng/$\mu$l internal standard PGB$_2$-d$_4$ (Cayman Chemicals) was added. The homogenate was left for 15 min on ice in the dark and then centrifuged at 4°C for 10 min at 130$g$ to remove precipitated proteins. The supernatant was transferred to a glass vial and acidified to pH 3.0 using 0.1 M HCL. Acidified extracts were applied to the pre-conditioned solid phase extraction cartridges (C18-E; Phenomenex) where lipid mediators were eluted with methyl formate. Extracts were dried under nitrogen, in the dark, and reconstituted in 100 $\mu$l 70/30:vol/vol ethanol/water.

## LC/ESI-MS/MS analysis

Lipid mediator analysis was carried out as described previously (Massey & Nicolaou, 2013). LC-MS/MS analyses were performed on a Waters Alliance 2695 HPLC pump coupled to an ESI triple quadrupole Quattro Ultima mass spectrometer (Waters). Chromatographic separation of the lipid mediators was carried out using a C18 Luna column (5 $\mu$m, 150 × 2.0 mm; Phenomenex). Chiral analysis of 9- and 13-HODE was carried out using a LUX cellulose-1 column (3 $\mu$m, 150 × 2.0 mm; Phenomenex). The metabolites were measured using multiple reaction monitoring using the following transitions: PGE$_2$: m/z 351>271, m/z 351>315, m/z 351>333; PGF$_{2\alpha}$: m/z 353>193, m/z 353>247, m/z 353>309; 9-HODE: m/z 295>171; 13-HODE: m/z 295>195. Quantitation was based on calibration lines constructed with commercially available standards (Cayman Chemicals).

## Protein over-expression vectors

The doxycycline-inducible vector pTRE3G (Clontech) was modified to incorporate an optimised 3XFLAG tag (MDYKDHDGDYKDHDI-DYKDDDDK) to generate the vector pTRE3G-FLAG. cDNA encoding the haem peroxidase domain of Pxt (aa38-798) was cloned into pTRE3G-FLAG allowing over-expression of N-terminally 3XFLAG-tagged proteins. Sequences were inspected using SignalP 4.1 to exclude signal peptide and pro-protein cleavage sites.

## Stable cell lines

HeLa cells were split to 50% confluency and transfected with 3 $\mu$g pCMV-Tet3G activator vector and 500 ng of each pTRE3G-FLAG expression vector using Lipofectamine LTX (Invitrogen) in serum-free medium. Transfected cells were incubated at 37°C with 5% CO$_2$ overnight. Stable transfectants were selected with 300 $\mu$g/ml neomycin (Sigma-Aldrich).

## Protein over-expression and purification

Stable cell lines and non-transfected HeLa cells (control) were grown to 50% confluency in T150 flasks (Corning) and FLAG-tagged protein expression induced with 5 µg/ml of doxycycline (Sigma-Aldrich) for 48 h. After 24 h, the medium was discarded and fresh medium with doxycycline was added. The cells were trypsinized using 1× trypsin–EDTA (Sigma-Aldrich) for 5 min and pelleted by centrifugation at 300$g$ for 5 min at 4°C. Pellets were washed with 1× PBS and lysed in 1× RIPA buffer (Sigma-Aldrich) containing protease inhibitors (Complete; Roche) on ice for 30 min. Lysates were cleared by centrifugation at 4900$g$ for 10 min at 4°C. FLAG-tagged proteins were purified by incubation with Anti-FLAG M2 monoclonal antibody (Sigma-Aldrich)–coated Protein G Dynabeads (Life Technologies). The beads were washed three times with 1× RIPA buffer (Sigma-Aldrich) and proteins eluted by two incubations with 20 µl elution buffer (100 µg/ml 3× FLAG peptide [Sigma-Aldrich], 100 mM Tris–HCl [pH 8.0], and 100 mM NaCl, 20% glycerol). Eluted protein was stored at –80°C until use.

## MPO assay

Protein levels of purified fusion proteins were normalised by Bradford assay (Bio-Rad) and confirmed by Western blotting. FLAG-tagged proteins were separated on 8% SDS–PAGE gels, transferred to polyvinylidene fluoride (PVDF) membrane (Millipore), and detected using mouse anti-FLAG M2 monoclonal antibody (Sigma-Aldrich) at 1:1,000, anti-mouse HRP-conjugated IgG (H+L) secondary antibody (Amersham) at 1:10,000, and SuperSignal West Pico Chemiluminescent Substrate (Thermo Fisher Scientific). To assay MPO activity, the COX Fluorescent Activity Assay Kit (Cayman Chemical) protocol was modified by substituting AA in the assay with 10 µl 5 mM hydrogen peroxide. Assays were monitored using a Fluoroskan (Ascent 1.6; Labsystems) with an excitation wavelength of 550 nm and an emission wavelength of 615 nm. Ascent research addition 1.2.3.1 program was used to analyse the data. Duplicate reactions containing MPO inhibitor 1 (Calbiochem) were also assayed. The COX standard internal control provided by the assay kit was used as a positive control.

## ChIP

ChIP was performed on mock and 9-(S)-HODE treated S2 cells (10 µM for 24 h) as described in Kwon et al (2016). Briefly, 10 million cells were fixed by addition of 1% formaldehyde for 15 min at 25°C, pelleted, and washed 3× in ice-cold 1× PBS before storage at –80°C. The cell pellets were resuspended in 150 µl ChIP Lysis Buffer (1% SDS, 10 mM EDTA) and sonicated for 20 min using a Diagenode Bioruptor sonicator (cycles of 30 s on/off, hi power). Samples were diluted 10-fold in ChIP dilution buffer. The samples were pre-cleared using Protein G–conjugated Dynabeads (Invitrogen) for 30 min at room temperature, followed by incubation with antibody-coated Protein G–conjugated Dynabeads (Invitrogen) for 3 h at 4°C. Immune complexes were recovered by magnetic selection and washed once with low-salt buffer (20 mM Tris [pH 8.1], 150 mM NaCl, 2 mM EDTA, 0.1% SDS, and 1% Triton X-100), once with high-salt buffer (20 mM Tris [pH 8.1], 500 mM NaCl, 2 mM EDTA, 0.1% SDS, and 1% Triton X-100), once with LiCl immune complex wash buffer (10 mM Tris [pH 8.1], 1 mM EDTA, 0.25 M LiCl, 1% IGEPAL CA-630, and 1% deoxycholic acid), and twice with TE buffer for 5 min each at room temperature. ChIP DNA was eluted using two washes of elution buffer (1% SDS and 0.1 M NaHCO$_3$) for 15 min at room temperature. Cross-links were reversed as described above and ChIP DNA purified using 1.8 volumes Agencourt AMPure XP beads (Beckman Coulter). Rabbit anti-dFOXO antibodies (Donovan & Marr, 2016) were used for immunoprecipitation.

Enrichment at known dFOXO targets was assayed by real-time PCR relative to control genomic loci using primers listed in the key resources table. Real-time PCR was performed on an Applied Biosystems Step One Plus real-time PCR machine. Reactions were performed using Absolute QPCR SYBR green ROX mix (AB-1162; Thermo Fisher Scientific). Primers used were: *foxo-control-f* AGCCCGCGAAGATACAAGAG; *foxo-control-r* AGCCACAAACAGCGACAGAA; *pdcd4-f* CCGTTGGGAGTCTCTCTCTC; *pdcd4-r* GCTCTCCCCACCTTCTCACC; *hsp26-f* GTGCGCCTGTATGAGTGAGA; *hsp26-r* GTGGGAGATTGCTGGCGTTA; *4EBP-f* CAAGAACCAGCCGGTTTGTC; *4EBP-r* GCTCTCTTCTCGCTCTTTCG. ChIP values in the presence and absence of 9-HODE at known FOXO targets and control genomic regions were calculated as a % of input using the Delta-Ct method. Input cycles were adjusted (Ct-6.64) to take into account that 1% of chromatin used in the ChIP was reserved for input. These ChIP values were then used to generate fold enrichment after addition of 9-HODE.

## Immunofluorescence microscopy

Immunofluorescence was performed on mock and 9-(S)-HODE–treated S2 cells (10 µM for 24 h). For experiments to investigate whether 9-(S)-HODE requires ROS to induce dFOXO nuclear localisation, S2 cells were simultaneously treated with 100 µg/ml N-acetyl-L-cysteine (NAC; Sigma-Aldrich). To test whether 9-(S)-HODE induces ROS, mock and 9-(S)-HODE–treated S2 cells (10 µM for 24 h) were stained using CellROX Green Reagent (Life Technologies) that allows oxidative stress to be detected in live cells. Insulin treatment (from bovine pancreas; Sigma-Aldrich) was performed on mock and 9-(S)-HODE–treated S2 cells (10 µM for 24 h) at 1 µg/ml for 90 min. FCS was added to cell suspension to a final concentration of 20% and the cells centrifuged onto glass slides using a Cytospin3 (Shandon) slide centrifuge for 5 min at 112$g$ on low acceleration and deceleration setting. The cells were fixed with 4% paraformaldehyde for 10 min at room temperature and washed with 1× PBS for 10 min before blocking overnight at 4°C in blocking buffer (1× PBS, 0.1% Tween-20, 0.1% FCS). Primary antibody incubation was performed in blocking buffer for 2 h followed by three washes in PBTw buffer (1× PBS, 0.1% Tween-20). Secondary antibody incubation was performed in PBTw at room temperature for 1 h using either Cy3-conjugated AffiniPure goat anti-rabbit antibody (Cat. no. 111-167-003; Jackson ImmunoResearch) or FITC-conjugated AffiniPure Donkey anti-mouse antibody (Cat. no. 715-095-151; Jackson ImmunoResearch) at 1:400 dilution. The samples were washed as above and mounted using Vectashield with DAPI (Vecta Laboratories). Primary antibodies used were rabbit anti-dFOXO (Donovan & Marr, 2016) (1:1,000) and mouse anti-pJNK (phospho-SAPK/JNK (Thr183Tyr185) MAb G9, CST, 1:100).

Fat bodies and salivary glands were dissected from *dFOXO-mCherry* third instar larvae. Tissue was incubated in insect-express medium containing L-glutamine (Lonza) with 10% FCS and with or without addition of 10 $\mu$M 9-(S)-HODE. Incubations ranged from 3 h to overnight (18 h) incubations. The samples were fixed in 4% paraformaldehyde for 15 min at room temperature, followed by three washes in 1× PBS and mounted using Vectashield with DAPI (Vecta Laboratories). dFOXO nuclear localisation was imaged using a Zeiss LSM780 confocal microscope. Control (mock-treated) and experimental samples (9-(S)-HODE–treated, 13-(S)-HODE–treated, and 9-(S)-HODE–treated + RNAi) were fixed and imaged in parallel using identical imaging settings. Images were exported as TIFF files and nuclear signals quantitated using ImageJ. Data represent mean and SD of at least 10 image panels per experiment (5–7 nuclei per panel).

## Western blotting

Mock and 9-(S)-HODE treatment of S2 cells was performed in six-well plates (three wells per condition) as described above. After treatment, the cells were resuspended and wells pooled then pelleted by centrifugation at 300$g$ for 5 min at 4°C in a 15-ml falcon tube. The media was removed and cells washed in 5 ml 1× PBS. Cell pellets were resuspended in 1× RIPA buffer (Sigma-Aldrich) containing protease inhibitors (Complete; Roche) and cells lysed by sonication on ice (30% power, for 30 s, four repeats) using a Vibra cell VC130 sonicator (Sonics). The lysate was cleared by centrifugation for 10 min at 21,000$g$ using a microcentrifuge. The supernatant was transferred to a new tube and protein concentration measured using 1× Bradford assay (Bio-Rad). 25 $\mu$g of extract from each induced protein line was ran on an 10% SDS–PAGE gel using an XCell SureLock Mini-Cell Electrophoresis System (Invitrogen) with Tris–glycine running buffer (25 mM Tris, 192 mM glycine, and 0.1% SDS) at 125 V for 2 h. Proteins were transferred to PVDF membrane (Bio-Rad) using an X Cell II Blot Module (Invitrogen) and Tris–glycine transfer buffer (12 mM Tris and 96 mM glycine) containing 20% methanol at 30 V for 2 h. Membrane was blocked overnight at 4°C in 1× Tris buffered saline with Tween-20 (TBST [50 mM Tris–Cl, pH 7.4, 150 mM NaCl, and 0.1% tween 20]) containing 5% dried skimmed milk powder. Primary antibody incubation was performed in blocking buffer at room temp for 1.5 h. Membranes were washed three times for 10 min each in TBST and secondary antibody incubation performed in TBST for 1 h at room temperature. Membranes were washed as above and the signal was detected following Supersignal West Pico Chemiluminescent Substrate (ECL substrate; Thermo Fisher Scientific) reaction for 5 min. The signal was visualised using Hyperfilm ECL (Amersham). Primary antibodies used were rabbit anti-dFOXO (Donovan & Marr, 2016) (1:1,000), mouse MAb E7 anti-tubulin (Developmental Studies Hybridoma Bank, 1:100), and mouse anti-pJNK (anti-phospho-SAPK/JNK (Thr183Tyr185) MAb G9, CST, 1:2,000). Secondary antibodies were anti-mouse and anti-rabbit HRP-conjugated IgG (H+L) secondary antibody (Amersham) at 1:10,000.

## Survival and infection assays

Natural lifespan and survival after infection was assayed using $Pxt^{f05258}$ and $w^{1118}$ flies. For each fly strain, five replicates of 50 male flies, in total 250 flies, were used per assay. Adult virgin males were collected, aged for 1 d, and placed in vials containing standard *Drosophila* medium. Vials were incubated at 25°C and survival recorded daily. The medium was changed regularly during the assays. Kaplan–Meier survival curves were produced using the SPSS statistics package. Survival in response to sterile wounding was tested. Flies were anaesthetised on a $CO_2$ pad and wounded between the A5 and A4 dorsal abdominal tergites using a fine-sterilised platinum needle that had been dipped in sterile 1× PBS. Survival after exposure to *B. bassiana* (CABI), a pathogenic fungal strain, was also assayed. *B. bassiana* was cultured on potato and carrot agar plates for 3 d when it reached peak sporulation. Flies from each genetic strain were anesthetised using $CO_2$. For the control group, for each replicate, the flies were placed onto a sterile potato and carrot agar plate and shaken gently over the plate for 10 s and then placed into a fresh vial. For the infected group, flies were treated in the same way, but shaken over the *B. bassiana* grown potato and carrot agar plate.

## Real-time PCR analysis of transcript levels

mRNA was isolated using a $\mu$MACS mRNA Isolation kit (Miltenyi Biotec). cDNA was generated by reverse transcription using Superscript II (Invitrogen) at 50°C. Real-time PCR was performed on an Applied Biosystems Step One Plus real-time PCR machine. Reactions were performed using Absolute QPCR SYBR green ROX mix (AB-1162; Thermo Fisher Scientific). Primers used are described below. *RpL32* transcript provided the endogenous control for normalisation. RNA was isolated from five third instar larvae of the appropriate genotype and media treatment. Data are mean and SD of three biological replicates. Primers for PCR were as follows

*RpL32-F* ATCCGCCACCAGTCGCATCGATATGCTAAG,
*RpL32-R* TCTTGAGAACGCAGGCGACCGTTGGGGTTG;
*Dpt_F* GCAGTTCACCATTGCCGTCGCCTTACTTTG,
*Dpt_R* TGAAGATTGAGTGGGTACTGCGGTGGTGGA;
*Drs_F* ATCAAGTACTTGTTCGCCCTCTTCGCTGTC,
*Drs_R* CTCGTTGTCCCAGACGGCACAGGGACCCTT;
*TepI_F* CTGAAGTCTCAGTCAGCCTGACTGGACCTT,
*TepI_R* CGTAATCGCCTTCTGTTAGCTTCGGAATGT;
*IMPPP_F* ACCGATGAGGCGGGCAACAC,
*IMPPP_R* GGCCACGCTGAATGTAGAAT;
*TotA_F* TGCTTTGCACTGCTGCTGATTAGTCCTCTA,
*TotA_R* TTTCACTCAAATATTAAAACAATATTAACC;
*Upd3_F* GGTCACCTACAAGATACTGC,
*Upd3_R* TCGCCTTGCACAGACTCTTA.

# Data Availability

Microarray datasets are available through Array Express with accession number E-MTAB-6253.

# Supplementary Information

# Acknowledgements

We thank Michael Marr for providing anti-dFOXO antibodies; Sebastian Grönke and Linda Partridge for providing dFOXO-mCherry and dFOXO delete fly strains; Yun Fan for providing JNK strains; and the Bloomington *Drosophila* Stock Center and Exelixis Collection at the Harvard Medical School for providing Pxt mutants.

## Author Contributions

SY Kwon: conceptualization, data curation, formal analysis, supervision, funding acquisition, investigation, methodology, project administration, and writing—original draft, review, and editing.
K Massey: data curation, investigation, and methodology.
MA Watson: formal analysis and investigation.
T Hussain: formal analysis and investigation.
G Volpe: formal analysis and investigation.
CD Buckley: supervision, funding acquisition, and writing—review and editing.
A Nicolaou: supervision, funding acquisition, methodology, project administration, and writing—review and editing.
P Badenhorst: conceptualization, formal analysis, supervision, funding acquisition, investigation, project administration, and writing—original draft, review, and editing.

## Conflict of Interest Statement

The authors declare that they have no conflict of interest.

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
