## [Reviewer comments · Life Science Alliance]

Life Science Alliance

Oxidised metabolites of the omega-6 fatty acid linoleic acid activate dFOXO.

So Kwon, Karen Massey, Mark Watson, Tayab Hussein, Giacomo Volpe, Christopher Buckley, Anna Nicolaou, and Paul Badenhurst

DOI: <https://doi.org/10.26508/lsa.201900356>

Corresponding author(s): Paul Badenhurst, University of Birmingham

Review Timeline:

Submission Date:	2019-02-21
Editorial Decision:	2019-03-25
Revision Received:	2019-12-13
Editorial Decision:	2019-12-17
Revision Received:	2019-12-20
Accepted:	2019-12-23

Scientific Editor: Andrea Leibfried

Transaction Report:

March 25, 2019

Re: Life Science Alliance manuscript #LSA-2019-00356-T

Dr. Paul W Badenhorst
University of Birmingham
Institute of Biomedical Research
Department of Anatomy
Edgbaston, West Mids B15 2TT
United Kingdom

Dear Dr. Badenhorst,

Thank you for submitting your manuscript entitled "Oxidised metabolites of the omega-6 fatty acid linoleic acid control insulin signaling by activating dFOXO" to Life Science Alliance. The manuscript was assessed by expert reviewers, whose comments are appended to this letter.

As you will see, the reviewers appreciate your work and support publication of a revised version of your manuscript in Life Science Alliance. We would thus like to invite you to submit a revised version of your work, addressing the comments and concerns of the reviewers. A few experiments are needed (see points 5, 8a, 14, 15, 17, 19, 21 of reviewer #3), but many concerns can get addressed by adding clarifications and by changes to data representation / the manuscript text. We'd be happy to discuss such revision in more detail with you once you've had the time to read all comments carefully.

Thank you for this interesting contribution to Life Science Alliance. We are looking forward to receiving your revised manuscript.

Sincerely,

B. MANUSCRIPT ORGANIZATION AND FORMATTING:

Reviewer #1 (Comments to the Authors (Required)):

Referee's report on the manuscript: « Oxidized metabolites of the omega-6 fatty acid linoleic acid control insulin signaling by activating dFOXO »

This manuscript by Kwon et al reports a thorough metabolic and genetic analysis in *Drosophila* to study the influence of saturated and polyunsaturated fatty acids on Insulin signalling. This topic is of evident relevance to diseases related to obesity and insulin resistance and the powerful approaches developed in this work illustrate again the value of the fly model to tackle in an unbiased way important biomedical questions. The work has multiple angles and has been excellently designed, performed and reported. The results look conclusive and robust and the mechanistic depth of the study is to be highlighted. I have hardly any criticism to formulate and congratulate the authors for this excellent work. I feel this manuscript should be published in its current form, maybe after a final language editing.

Reviewer #2 (Comments to the Authors (Required)):

Kwon et al. have developed a *Drosophila* model to study the effects of dietary fats on inflammation status. As read-out of inflammation status they score formation of so-called melanotic tumors, formed by the aggregation of lamellocytes in response to inflammation. The authors monitored the effects of changing the balance of dietary fatty acids on inflammation and insulin signaling. Their analysis revealed that the balance between SFA and PUFA modulates the level of inflammation. In particular, omega-6 PUFAs enhance inflammation, whereas omega-3 PUFAs suppress inflammation. The authors use a lipidomics approach to establish that omega-6 PUFA enhances inflammation through the production of 9-HODE. To analyze the underlying molecular pathway, the authors performed a transcriptomics analysis that the FOXO transcription factor is the effector of 9-HODE. 9-HODE activates FOXO via the JNK signaling pathway, counteracting insulin signaling that normally down-regulates FOXO.

This is an interesting, well-executed study that identifies (linoleic acid-derived) 9-HODE as a novel mediator of pro-inflammatory signaling via FOXO. In my opinion this work should be of general interest to the readership of life science alliance. I have only a few minor comments.

- 1) I don't understand the quantification of the ChIP experiments. I am always suspicious of "relative ChIPs". In my opinion they should be presented as % input (with control regions and different conditions).
- 2) The FOXO blots are not great. How a time-course? dose-response?
- 3) Could the authors include some analysis where insulin is used as a control to down-regulate FOXO and in combination with 9-HODE? This would give a feel for the full range of FOXO regulation.
- 4) I might have missed-it, but what are the effects of FOXO over-expression or expression of a mutant constitutively active FOXO on inflammation? Can these experiments be performed?
- 5) The paper would be more generally accessible with a better introduction and a discussion that explains the nice model in Fig. 5L. Also for the non-experts some of the lipidomics parts in the results are hard-going.

Reviewer #3 (Comments to the Authors (Required)):

The authors investigate the causal links between obesity, systemic inflammation, and the insulin/dFoxo signaling, focusing on the consequences of high fat-diet rich in saturated or polyunsaturated fatty acids and using a *Drosophila* model of blood cancer. The authors successfully identified omega-6 derived metabolites that increased in the

cancer condition using lipidomic. Convincing data are also provided that in cell culture, linoleic acid-derived lipid mediator 9(S)HODE causes nuclear localization of FOXO, a negative regulator of the insulin/IGF signaling (IIS), and that disruption of the JNK prevents this effect. The authors also provide strong synergism between elements of these pathways and the tumor condition, suggesting that FOXO, JNK/Bsk, and linoleic acid-derived lipid metabolites and the Pxt enzyme contribute to the blood tumors induced by aberrant activation of the JAK/STAT pathway. Collectively, the manuscript provides interesting new findings implicating omega 6 PUFA in insulin resistance/sensitivity through regulation of JNK/Bsk-mediated control of FOXO that together may contribute to a better understanding of the link between inflammation and insulin resistance.

Specific comments and suggestions:

1. The authors need to better justify the use of the model and explain some of the statements. For example, they state that: "Lamellocytes are normally rare but, in response to inflammatory stimuli, differentiate from plasmatocytes and aggregate to produce inflammatory nodules called "melanotic tumors". Ref. 27. This reference refers to melanized masses by the crystal cells. Not all melanotic masses are formed by tumor cells and most are formed by crystal cells and not all include lamellocytes. This part is confusing
2. Please provide evidence that the melanotic masses are inflammatory nodules and also provide more informative evidence that hopTum blood masses are a good readout of inflammation.
3. What markers of inflammation are expressed in these mutants?
4. Please describe briefly the evidence that: "Activation of the JAK/STAT pathway in these mutants provides the trigger that initiates inflammation and the production of melanotic tumors (Fig 1A and B)."
5. The omega-3 PUFA significantly altered tumor incidence. It is not shown if it also affects inflammation as this aspect of the mutant phenotype is not characterized or qualified.
6. Please describe when the diet with the SFA and PUFA are introduced (in larvae or adult stage) and how long the animals were in such diet before the analyses.
7. The suppressive effect of ALA is very interesting! but not followed up ...
8. To support that the Pxt mutant and the inhibitor mediates the effect of LA in enhancing tumor mass, further direct evidence is needed.
 - a. Is the level of 9(S)HODE altered in Pxt mutants? Can the tumor suppressive effect of the inhibitor compound and/or Pxt mutation be rescued by feeding the animals with 9(S)HODE?
9. The data in cell culture are very intriguing but totally disconnected with the data of tumorigenesis and 'inflammation'.
10. Given that the endogenous 9(S)HODE is the predominant lipid species already in non-inflammatory condition and the effect of ALA does not change this lipid species, further evidence is needed to connect the described effect with the endogenous role of 9(S)HODE in inflammation.
11. A large part of tumorigenesis in hopTum appears to be independent of 9(S)HODE, suggesting that if inflammation contributes to tumorigenesis, it may not be the main mechanism in this tumor type.
12. FOXO could act in tumorigenesis of hopTum without being a 'mediator' of the effects of 9(S)HODE. More direct and informative evidence must be provided. Thus far this is an inference from unrelated data in cell culture and in vivo.

13. The relationship between Jnk/Bsk and nuclear FOXO is of interest but how this is connected to the other part of the paper is not clear.
14. bsk-RNAi and Bsk-DN can reduce nuclear translocation of FOXO stimulated by 9(S)HODE in fat cells or does it reduce FOXO stabilization? What we can see is increased FOXO levels but we do not see cytoplasmic-nuclear translocation.
15. This experiment requires a further control of the effects of bsk without supplementation of 9(S)HODE. What concentration of 9(S)HODE was used?
16. In Fig 5H. In mock dFOXO is nuclear but show a weaker expression. Similarly, in Fig. EV4A, there is no expression of FOXO in the cytoplasm or expression at all but expression is seen in the nucleus in 9(S)HOSE cells. Is the lipid regulating FOXO at the transcriptional level? This is not the typical nuclear-cytoplasmic switch seen in S2 cells upon insulin stimulation.
17. Fig. 5F It seems that Bsk inhibits Foxo-mCherry. One cannot see cytoplasmic FOXO-mCherry that would be expected in the fat body cells upon nuclear-cytoplasmic translocation of the Foxo protein.
These experiments need further controls.
18. A better explanation of the relevance of these results in the context of inflammation and related with the hopTum should be provided.
19. The authors propose that control of dFOXO by 9-(S) HODE is mediated via the JNK pathway. This statement needs to be reinforced by direct evidence using double mutants and rescue to provide support that
- "oxidation of the omega-6 PUFA LA generates 9-(S)-HODE, in turn activating JNK to trigger elevated nuclear dFOXO and control expression of FOXO targets"
FOXO targets are induced in S2 cells stimulated with 9(S)HODE, but it needs to be validated in vivo in a well-characterized inflammation context.
20. It is intriguing that no ROS is associated with hopTum but still the mutation is used as a readout of inflammation
21. Page 445
"By driving constitutively elevated FOXO nuclear levels, 9-(S)-HODE has the potential to override IIS signaling inputs and provides a molecular mechanism by which insulin resistance could be established (Fig 5L)."
Given that insulin resistance is well characterized condition in flies, and is specifically associated with dwarf larvae and/or animals with high circulating levels of glucose, and high levels of Foxo target gene expression (widely used are the Foxo target genes InR and 4E-BP), the authors needs to provide evidence that induced the animals are insulin resistance.
22. The authors discuss extensively about obesity and inflammation and insulin resistance. However, are larvae with hopTum obese? How is obesity related to the experiments discussed here?
For example, the authors discuss about "Obesity coupled with low-grade inflammation is closely associated with the development of insulin resistance, and is speculated to require activation of stress-activated kinases like JNK". The two papers are related to work done in mice. Please add a paper supporting this statement in flies in connection to the mutant hopTum used here

Minor comments:

1. The title highlights parts of the study but it does not describe the main focus of the study. The current title appears to imply that the authors have characterized insulin signaling and function of Foxo, but the evidence is mainly on nuclear localization or increased nuclear stability and the expression profile in S2 cells.

Response to reviewers

We would like to express our gratitude to the reviewers for both their positive and their critical comments. We have addressed many of their concerns and feel that the manuscript has improved considerably as a result. Most importantly, we have now performed additional experiments and made text changes that we feel help to focus the manuscript and provide further supporting evidence for our model.

Below, please, find a point-by-point reply to the reviewer's comments (reviewer comments in italics).

Referee 1:

This manuscript by Kwon et al reports a thorough metabolic and genetic analysis in drosophila to study the influence of saturated and polyunsaturated fatty acids on Insulin signalling. This topic is of evident relevance to diseases related to obesity and insulin resistance and the powerful approaches developed in this work illustrate again the value of the fly model to tackle in an unbiased way important biomedical questions. The work has multiple angles and has been excellently designed, performed and reported. The results looks conclusive and robust and the mechanistic depth of the study is to be highlighted. I have hardly any criticism to formulate and congratulate the authors for this excellent work. I feel this manuscript should be published in its current form, maybe after a final language editing.

We thank the reviewer for their expression of support and confidence in our manuscript and especially their emphasis of the power and utility of fly models to investigate biomedical questions. Thank you!

Referee 2:

This is an interesting, well-executed study that identifies (linoleic acid-derived) 9-HODE as a novel mediator of pro-inflammatory signaling via FOXO. In my opinion this work should be of general interest to the readership of life science alliance. I have only a few minor comments.

Likewise we thank this reviewer for their expression of support for our manuscript and their confidence in the general interest of our manuscript. We thank them for their helpful suggestions which have helped to improve the quality of the manuscript. To address their comments on the manuscript we have performed additional experiments and made changes to the text as detailed below. We hope that these changes address the remaining concerns of this reviewer.

1) I don't understand the quantification of the ChIP experiments. I am always suspicious of "relative ChIPs". In my opinion they should be presented as % input (with control regions and different conditions).

Indeed, we agree with the reviewer. This was the method that was followed in our ChIP analysis. ChIP values in the presence and absence of 9-HODE at known FOXO-targets and control regions were calculated as a % of input using the Delta-Ct method. Input cycles were adjusted to take into account that 1% of chromatin used in the ChIP was reserved for input. These ChIP values were then used to generate fold enrichment after addition of 9-HODE (data plotted in Fig 4G). We have clarified the analysis method utilized in the methods section and also that Cont in Fig 4G represents control genomic region that is not anticipated to bind dFOXO.

2) The FOXO blots are not great. How a time-course? dose-response?

We agree with this reviewer that the quality of the previous Western Blot was below par. To address this we have performed additional Western Blots using anti-FOXO antibodies. This data is added as a new Figure 4C, and includes a time-course analysis of the response of FOXO to 9-HODE stimulation. This helps to address concerns raised by reviewer 3 (see **points 14,16,17** below) concerning the nature of the response of FOXO to 9-HODE, whether post-translational or transcriptional, and demonstrates rapid elevation of FOXO protein levels with 3 hours of stimulation. We hope that the quality of the new western and the addition of a time-course analysis satisfies addresses any outstanding concerns of the reviewer.

3) Could the authors include some analysis where insulin is used as a control to down-regulate FOXO and in combination with 9-HODE? This would give a feel for the full range of FOXO regulation.

This was included in **Fig 4J,K** showing that Insulin action can be overcome by 9-HODE.

4) I might have missed-it, but what are the effects of FOXO over-expression or expression of a mutant constitutively active FOXO on inflammation? Can these experiments be performed?

This is a good point. We did not specifically address this in the initial draft of the manuscript, but had analysed the effect of ectopic expression of WT dFOXO in macrophages on inflammation. We assayed the effect of ectopically-expressing WT dFOXO using the plasmacyte-expressed *Peroxidasin-GAL4* driver. A panel of *UAS-dFOXO* lines obtained from the Bloomington stock center were assayed for plasmacyte (Pxn+ve) to lamellocyte (L1+ve) fate switching. While an increase in lamellocyte numbers were observed in a single *UASdFOXO* line (*UAS-foxo-P*) this was not consistently observed with other lines tested. This data is included in the rebuttal below, but has not been included in the main manuscript.

It is worth noting that a similar experiment was discussed tangentially in Owusu-Ansah and Banerjee (2009) (PMID: 19727075) where over-expression of WT dFOXO in the lymph gland was reported as not inducing excess lamellocyte production. It is important to stress however that ectopically-expressed WT dFOXO would likely still be subject to the same inhibitory regulatory inputs (nuclear export/degradation) as dFOXO expressed from the endogenous locus. It is, thus, perhaps unsurprising that ectopically-expressed WT dFOXO does not induce lamellocyte differentiation. As the reviewer mentions – the more revealing experiment would be the over-expression of constitutively-active dFOXO. However, we currently do not know the identity of post-translational modification sites on dFOXO that are induced in response to 9-HODE treatment to activate dFOXO. A long-term goal is to use mass spectrometry to map post-translational modification sites on dFOXO following 9-HODE treatment. We plan to use this information to then engineer constitutively active variants of

dFOXO and assay effect of ectopic expression of these on inflammation and lamellocyte differentiation and ultimately on insulin responses.

5) The paper would be more generally accessible with a better introduction and a discussion that explains the nice model in Fig. 5L. Also for the non-experts some of the lipidomics parts in the results are hard-going.

We apologize for brevity in parts of the manuscript. The manuscript was initially formatted as a Scientific Report for EMBO Reports and transferred unedited to LSA. The Scientific Report format is an abbreviated format that inevitably curtails the detail in which some sections can be presented. We have addressed this in the draft by expanding our introduction to the fly model utilized (first paragraph of the results). This expanded section also partly address some concerns of reviewer 3 see **points 1-4** below). We have also edited and expanded paragraph 4 of the discussion to better describe the model presented in Fig. 5L and to link this model to our observed results and potential future experiments.

We do appreciate and understand the concerns of the reviewer regarding the lipidomics data sections, but beg their understanding regarding the terminologies and data presented in this section. In these sections we have attempted to strike the difficult balance between on the one hand simplifying the lipidomics terminology and data so that it is accessible to a general audience, while at the same time retaining sufficient detail and rigour so as to be adequately judged by specialist lipidomics experts. If this balance has swayed in favour of specialist lipidomics we can only apologize as we felt it was necessary to include sufficient detail to allow an accurate assessment of the results presented. This is especially so given the implications of some of our results for previous publications that have inferred (without lipidomics evidence) the existence of certain lipid mediator species in flies.

Referee 3:

The authors investigate the causal links between obesity, systemic inflammation, and the insulin/dFoxo signaling, focusing on the consequences of high fat-diet rich in saturated or polyunsaturated fatty acids and using a Drosophila model of blood cancer. The authors successfully identified omega-6 derived metabolites that increased in the cancer condition using lipidomic. Convincing data are also provided that in cell culture, linoleic acid-derived lipid mediator 9(S)HODE causes nuclear localization of FOXO, a negative regulator of the insulin/IGF signaling (IIS), and that disruption of the JNK prevents this effect. The authors also provide strong synergism between elements of these pathways and the tumor condition, suggesting that FOXO, JNK/Bsk, and linoleic acid-derived lipid metabolites and the Pxt enzyme contribute to the blood tumors induced by aberrant activation of the JAK/STAT pathway. Collectively, the manuscript provides interesting new findings implicating omega 6 PUFA in insulin resistance/sensitivity through regulation of JNK/Bsk-mediated control of FOXO that together may be may contribute to a better understanding of the link between inflammation and insulin resistance.

We thank the reviewer for their support for our manuscript and confidence that data presented here provides a new component to understanding the link between inflammation and insulin resistance. We thank the reviewer for their many helpful suggestions that have helped to focus and improve the manuscript. Below we address in detail each point raised by this reviewer.

1. The authors need to better justify the use of the model and explain some of the statements. For example, they state that: "Lamellocytes are normally rare but, in response to inflammatory stimuli, differentiate from plasmatocytes and aggregate to produce inflammatory nodules called "melanotic tumors". Ref. 27. This reference refers to melanized

masses by the crystal cells. Not all melanotic masses are formed by tumor cells and most are formed by crystal cells and not all include lamellocytes. This part is confusing

Indeed, it is certainly the case that melanization is traditionally considered the response of crystal cells and that some melanization reactions result from the reaction of crystal cells with no involvement of lamellocytes. To respond, we and others have shown previously (Kwon et al. (2008)) that hop[Tum] mutants, which display melanotic tumours, have elevated lamellocyte numbers and that tumor incidence is correlated with the percentage of plasmatocytes that differentiate into lamellocytes (by varying growth temperature and increasing the degree of STAT activation). Moreover, genetic complementation experiments show that mutations that reduce lamellocyte production also reduce melanotic tumour formation indicating a close causal relationship lamellocyte production and melanotic tumour formation. In this manuscript we have crossed the fluorescent lamellocyte reporter MSNF9mo-mCherry into the hop[Tum] mutant background to label lamellocytes, dissected melanotic tumours from adults and shown that the features that we are scoring as melanotic tumours in our experiments are composed of fluorescently labelled lamellocytes (**Fig. 1B**). These data do not exclude the possibility that crystal cells are recruited into lamellocyte aggregates and contribute to melanization. Certainly a feature of most inflammatory responses is the activation and local recruitment of leukocytes, usually of more than a single type, often following an ordered hierarchy of accumulation. As such there is no reason to expect that crystal cells would not also be recruited to melanotic masses. Unpublished data from our lab, however, would argue that melanization in melanotic tumours is not to sole preserve of crystal cells. mRNA analysis (Kwon et al. (2008)) and unpublished CAGE data (see below) indicate that lamellocytes express high-levels of prophenoloxidase activity (Dox-A3/PPO3). This concurs with publications from the Lee lab showing that Dox-A3/PPO3 mutants reduce melanization in hop[Tum] mutants (PMID: 18852525) and data from the Meister laboratory that crystal cells are actually rare in in hop[Tum] mutants (PMID: 11161576), suggesting that lamellocytes also melanize.

2. Please provide evidence that the melanotic masses are inflammatory nodules and also provide more informative evidence that hopTum blood masses are a good readout of inflammation.

As detailed in point 1 above, we have crossed the fluorescent lamellocyte reporter MSNF9mo-mCherry into the hop[Tum] mutant background to label lamellocytes, dissected melanotic tumours from adults and shown that the masses that we are scoring as melanotic “tumours” in our experiments are composed of fluorescently-labelled lamellocytes (**Fig. 1B**).

In mammals, the gross morphological cognates of inflammation are Redness, Swelling, Pain, and Heat which, in turn, reflect cellular hallmarks of excess **proliferation**, **activation** and local **accumulation** of leukocytes. Inflammation progresses through stages of initiation in which leukocytes recognize threats and are activated to release substances including vasodilatory histamines, cytokines and pro-inflammatory lipid mediators, that amplify and propagate initial reactions to increase proliferation, activation and accumulation of additional leukocytes to deal with the threat. Once the initial threat is dealt with, this is then followed by a process of resolution where anti-inflammatory mediator production limits leukocyte activation and infiltration to attenuate the inflammatory response and prevent chronic tissue damage.

Previously published data from many labs including our own show that lamellocytes are an **activated** hemocyte type that differentiate from plasmatocytes in response to wounding and infestation. Here we show that these **accumulate** in melanotic masses. As such, by definition melanotic masses/tumours reflect an inflammatory response as they are readouts of the **proliferation, activation** and aberrant **accumulation** of fly leukocytes. In addition, as indicated in a new **Fig 1C**, the melanotic tumour (hop[Tum]) background also exhibits the elevated expression of known humoral markers of inflammation in flies – target genes Dipteracin, Drosomycin, IMPPP, Turandot A, Tepl and the cytokine Upd3. Significantly, these are further elevated following stimulation with 9-HODE which also increases melanotic tumour incidence, further confirming that melanotic masses are a good readout of inflammation.

Moreover as in “normal” inflammatory responses where inflammation transitions through discrete stages of initiation to either propagation and/or resolution, with the balance between propagation and resolution being controlled by the production of pro- and anti-inflammatory mediators we would argue that JAK/STAT activation initiates an inflammatory response in hemocytes. However the severity of the final response – whether it is amplified and propagated or resolves - can be modulated by manipulation of the levels of pro- and anti-inflammatory mediators allowing melanotic tumour number to be used as an assay for pro- and anti-inflammatory mediators.

3. What markers of inflammation are expressed in these mutants?

As detailed in point 2 above, Dipteracin, Drosomycin, IMPPP, Turandot A, Tepl and the cytokine Upd3 have been analysed in a new figure (**Fig. 1C**).

4. Please describe briefly the evidence that: "Activation of the JAK/STAT pathway in these mutants provides the trigger that initiates inflammation and the production of melanotic tumors (Fig 1A and B)."

See point 2 above. There is a substantial body of evidence detailing JAK/STAT activation in the hop[Tum] mutant strain and associated increases in hemocyte number and hemocyte differentiation/activation to generate lamellocytes. **This evidence is detailed in an expanded first paragraph of the results section** but data supporting a trajectory from the hop[Tum] mutation to STAT activation, lamellocyte differentiation and melanotic tumour production includes key publications from the Dearolf, Perrimon Darnell and many other laboratories (PMID: 8479437; PMID: 7796812; PMID: 7729418; PMID: 8608595; PMID: 8608596; PMID: 11161576). This is borne out by studies in which deletion of either STAT (PMID: 8608595) or direct targets of the STAT transcription factor (PMID: 20168330) suppress lamellocyte differentiation and melanotic tumour formation in hop[Tum] mutants. The hallmarks of an inflammatory response are the **excess proliferation, activation and local accumulation of leukocytes**. As STAT activation in the hop[Tum] mutant drives hemocyte **proliferation, activation** to form lamellocytes, and **local accumulation** to form melanotic tumours/nodules we would argue that STAT activation in the hop[Tum] mutant background initiates an inflammatory response in fly hemocytes, a consequence of which is the initiation of melanotic tumour production.

5. The omega-3 PUFA significantly altered tumor incidence. It is not shown if it also affects inflammation as this aspect of the mutant phenotype is not characterized or qualified.

Indeed, as described in the discussion, we speculate that omega-3 PUFA ALA acts by the production of 3-series mediators that are structurally similar to HODEs called HOTrEs (9- and 13-HOTrE). Conclusive analysis of this requires the assay and detection of these mediators in flies. In turn this demands the establishment of similar LC-MS/MS assays that have been used to detect 9-HODE and 13-HODE. Until recently this has been limited by the availability of 9- and 13-HOTrE standards for analysis. We are clearly keen to progress this work but the establishment of such assays from scratch is not trivial and we feel is beyond

the scope of the existing manuscript. Certainly it is an area of work we wish to continue as, as described in the 5th paragraph of the discussion, we believe the 3-series mediators would provide an avenue of therapeutic intervention to prevent/overcome any effects of omega-6 PUFA on insulin resistance. Preliminary data indicate that co-treatment of S2 cells with 9-HODE and 9-HOTrE prevents dFOXO nuclear accumulation, but the absence of reliable lipidomics assays to interrogate HOTrE abundance and variants make these results too preliminary for publication at this stage.

6. Please describe when the diet with the SFA and PUFA are introduced (in larvae or adult stage) and how long the animals were in such diet before the analyses.

This is a good point. FAs and HODEs were added to fly media prior to pouring into vials, and *Drosophila* raised in the constant presence of supplements (from egg through larva to adult). We have added a line detailing this to the relevant section of the Methods (*Drosophila* larval fatty acid supplementation).

7. The suppressive effect of ALA is very interesting! but not followed up ...

See point 5 above. This is indeed being followed up but has been limited by the availability of good lipidomic assays (LC-MS/MS) for the 3-series lipid mediators that we speculate are generated by ALA. As described above, we have performed some supplementation experiments using HOTrEs but, in the absence of robust assays for the presence of HOTrEs and associated experiments to determine chiral enantiomer ratios and relative levels of 9- and 13-HOTrE versus 9- and 13-HODE we feel that such experiments, while interesting, lack secure biological foundations. We are clearly keen to progress this work but it will require additional funding to enable the establishment of the assays to provide a strong biological foundation for further investigation – in particular the signaling pathways that are regulated by HOTrEs and how they interact with dFOXO.

8. To support that the Pxt mutant and the inhibitor mediates the effect of LA in enhancing tumor mass, further direct evidence is needed.

a. Is the level of 9(S)HODE altered in Pxt mutants? Can the tumor suppressive effect of the inhibitor compound and/or Pxt mutation be rescued by feeding the animals with 9(S)HODE?

Indeed we show in **a new Fig 2I** that 9(S)-HODE enhances melanotic tumour production in hop[Tum] mutants and, unlike stimulation of melanotic tumour production by LA supplementation in **Fig 2G and 2H**, that reduction in Pxt fails to reduce increased melanotic tumour number stimulated by 9(S)-HODE. This places 9(S)-HODE downstream of Pxt.

9. The data in cell culture are very intriguing but totally disconnected with the data of tumorigenesis and 'inflammation'.

We would argue that cell culture provides a simplified system in which to dissect the core mechanism by which 9-(S)-HODE signals. We have used cell-culture to identify FOXO activated by JNK as transducers of 9-(S)-HODE. In turn we have utilized this information to show that suppression of FOXO and JNK suppresses formation of melanotic tumour masses and inflammation, indicating that the use of cell cultures is actually a useful strategy to illuminate inflammation-relevant signaling mechanisms.

10. Given that the endogenous 9(S)HODE is the predominant lipid species already in non-inflammatory condition and the effect of ALA does not change this lipid species, further evidence is needed to connect the described effect with the endogenous role of 9(S)HODE in inflammation.

Indeed, as **described in point 5 and 7 above**, we speculate that omega-3 PUFA ALA acts by the production of 3-series mediators called HOTrEs (9- and 13-HOTrE) not by changes in the levels of 9-(S)-HODE. As observed with eicosanoid lipid mediators, the anti-inflammatory effects of pro-resolution mediators is achieved by competition with pro-inflammatory lipid mediators, not reduction in the levels of these mediators. It is the ratio between levels of the pro-inflammatory and anti-inflammatory lipid mediators (not the

absolute level of either) that controls the transition between propagation and resolution of inflammation. The key future experiment is to determine the relative ratio of 9-(S)-HODE to 9-(S)-HOTrE in conditions of ALA supplementation.

11. A large part of tumorigenesis in hopTum appears to be independent of 9(S)HODE, suggesting that if inflammation contributes to tumorigenesis, it may not be the main mechanism in this tumor type.

See **point 2 and 4 above**. We would argue that as in “normal” inflammatory responses where inflammation transitions through discrete stages of initiation to either propagation and/or resolution that JAK/STAT activation initiates an inflammatory response in hemocytes that leads to the production of melanotic tumours. However the severity of the final response – whether it is amplified/propagated or resolves - can be modulated by manipulation of the levels of pro- and anti-inflammatory mediators. In the model we fully expect JAK/STAT activation in fly leukocytes to trigger the initiation of melanotic tumour formation. What we assay is whether the severity of this response can be increased by increasing 9-(S)-HODE levels or concomitantly decreased by pro-resolution mediators derived from ALA. However the initiating inflammatory stimulus remains irrespective of levels of lipid mediators.

12. FOXO could act in tumorigenesis of hopTum without being a 'mediator' of the effects of 9(S)HODE. More direct and informative evidence must be provided. Thus far this is an inference from unrelated data in cell culture and in vivo.

I guess the question is whether it is reasonable to conclude on the basis of data showing:

- i) 9-(S)-HODE regulates dFOXO targets,
- ii) 9-(S)-HODE regulates dFOXO nuclear localization,
- iii) 9-(S)-HODE triggers dFOXO binding to dFOXO targets,
- iv) 9-(S)-HODE activates JNK,
- iv) JNK is required for 9-(S)-HODE to control dFOXO nuclear localization,
- vi) 9-(S)-HODE increases melanotic tumours in hop[Tum] mutants,
- vii) dFOXO mutants decrease melanotic tumours in hop[Tum] mutants, and
- viii) JNK mutants decrease melanotic tumours in hop[Tum] mutants,

that FOXO can mediate the effects of 9(S)-HODE. Of course anything is possible but, on balance, I tend to go with William of Ockham – “*Non sunt multiplicanda entia sine necessitate.*” The simplest solution most consistent with the data is usually likely to be the correct one.

13. The relationship between Jnk/Bsk and nuclear FOXO is of interest but how this is connected to the other part of the paper is not clear.

We respectfully disagree. This gets to the heart of how 9-(S)-HODE signals are transmitted. As summarized in Fig. 5L and stated in **point 12** above, our data show that 9-(S)-HODE activates JNK in turn driving nuclear accumulation of dFOXO that controls expression dFOXO targets. In addition we show in **Fig. 5A** that both JNK and dFOXO are required for melanotic tumour production, connecting them to the rest of the paper.

14. bsk-RNAi and Bsk-DN can reduce nuclear translocation of FOXO stimulated by 9(S)HODE in fat cells or does it reduce FOXO stabilization? What we can see is increased FOXO levels but we do not see cytoplasmic-nuclear translocation.

The short answer is we see both. We have added a **new Figure S4C and S4D** where we examine cytoplasmic and nuclear dFOXO-mCherry signals both in control untreated fat body, in untreated bsk-RNAi and bsk-DN cells and following 9-(S)-HODE treatment. As shown in **Fig S4D**, cytoplasmic dFOXO-mCherry signal decreases following 9-(S)-HODE treatment with concomitant increases in nuclear dFOXO-mCherry signal. However, as demonstrated in Fig. 4C and Fig 4D, the total levels of dFOXO (assayed by IF and Western blots) are also

increased by 9-(S)-HODE suggesting that dFOXO stability is also positively regulated by 9-(S)-HODE. There are precedents for this in previously published work showing that regulation of FOXO proteolysis controls abundance and transcriptional activity (PMID: 17276341; PMID: 26154057). An important result of this analysis is that Bsk inhibition either using RNAi or Bsk-DN constructs does not decrease cytoplasmic dFOXO-mCherry signal in the absence of 9-(S)-HODE treatment (new data **Figure S4C**).

15. This experiment requires a further control of the effects of bsk without supplementation of 9(S)HODE. What concentration of 9(S)HODE was used?

Supplementation experiments in whole larvae and adults were performed using 1 μ M 9-(S)-HODE. A line has been added to the Materials and Methods “9-(S)-HODE was supplemented to 1 μ M final concentration in media to assay effects on melanotic tumour formation”.

16. In Fig 5H. In mock dFOXO is nuclear but show a weaker expression. Similarly, in Fig. EV4A, there is no expression of FOXO in the cytoplasm or expression at all but expression is seen in the nucleus in 9(S)HOSE cells. Is the lipid regulating FOXO at the transcriptional level? This is not the typical nuclear-cytoplasmic switch seen in S2 cells upon insulin stimulation.

The comment is related to **point 14 above**. To summarize: we do not believe 9-(S)-HODE regulates dFOXO at the transcriptional level. Rather we observe both cytoplasmic to nuclear translocation and elevated dFOXO protein levels indicating additional control of protein stability. The time-frame of these increases (from Western within 3hr **Fig 4C** and 10 minutes by live imaging **Fig 4F**) make it unlikely that control is at the level of transcription.

17. Fig. 5F It seems that Bsk inhibits Foxo-mCherry. One cannot see cytoplasmic FOXO-mCherry that would be expected in the fat body cells upon nuclear-cytoplasmic translocation of the Foxo protein.

These experiments need further controls.

Indeed, as indicated in responses to **point 14 and point 16 above** this has been addressed by the addition of appropriate analyses in **a new Figure S4C and S4D**

18. A better explanation of the relevance of these results in the context of inflammation and related with the hopTum should be provided.

As discussed in **points 2, 3 and 4 above** we have expanded the introduction to the model as well as added additional data, including inflammatory markers expression (**Fig. 1C**) that we feel place both these results and the hop[Tum] model in the context of inflammation.

19. The authors propose that control of dFOXO by 9-(S) HODE is mediated via the JNK pathway. This statement needs to be reinforced by direct evidence using double mutants and rescue to provide support that "oxidation of the omega-6 PUFA LA generates 9-(S)-HODE, in turn activating JNK to trigger elevated nuclear dFOXO and control expression of FOXO targets" FOXO targets are induced in S2 cells stimulated with 9(S)HODE, but it needs to be validated in vivo in a well-characterized inflammation context.

We were not entirely clear the specific experiment that was required by the reviewer. However, we have established that FOXO and JNK are activated in response to 9-(S)-HODE, that inflammatory tumour production requires both FOXO and JNK, that Pxt which is involved in oxidation of LA to form 9-(S)-HODE is required for inflammatory tumour production but that its requirement can be rescued/by-passed by the addition of 9-(S)-HODE, and that addition of 9-(S)-HODE triggers not only inflammatory tumour production but also elevated expression of inflammation markers (**Fig. 1C**). It is unclear to us how double mutants in JNK and FOXO would reinforce this data given that we have already shown both to be required. In response to the second part of this comment we have shown that addition of 9-(S)-HODE triggers not only inflammatory tumour production but also elevated expression of inflammation markers (**Fig. 1C**). We should also stress that data in **Fig S2**

specifically assays the effect of Pxt (MPO) mutants on model inflammatory responses including sterile wounding and fungal infection. This data demonstrate that Pxt (MPO) and by extension 9-(S)-HODE is required to mount model inflammatory responses.

20. It is intriguing that no ROS is associated with hopTum but still the mutation is used as a readout of inflammation

We have no evidence to suggest this and, like the reviewer, would find it unlikely that there is no ROS associated with the hop[Tum] mutant. Quite the contrary, as summarized in Fig. 5L we would suggest that initiation of inflammation drives the generation of ROS that together with MPO oxidises LA to generate 9-(S)-HODE. To clarify, **we tested ROS in S2 cells Fig. 5J to exclude the possibility that 9-(S)-HODE causes ROS production.** It has been reported previously that ROS activates JNK. To exclude the possibility that 9-(S)-HODE triggers ROS that in turn activates JNK we tested ROS production in S2 cells after 9-(S)-HODE addition. We do not observe elevated ROS after 9-(S)-HODE treatment. On the contrary we believe that ROS lies upstream of 9-(S)-HODE. In fact ROS (H2O2) is a substrate that MPO utilizes to generate highly oxidizing hypohalous acids that mediate lipid peroxidation of LA to HODE.

21. Page 445 "By driving constitutively elevated FOXO nuclear levels, 9-(S)-HODE has the potential to override IIS signaling inputs and provides a molecular mechanism by which insulin resistance could be established (Fig 5L)." Given that insulin resistance is well characterized condition in flies, and is specifically associated with dwarf larvae and/or animals with high circulating levels of glucose, and high levels of Foxo target gene expression (widely used are the Foxo target genes InR and 4E-BP), the authors needs to provide evidence that induced the animals are insulin resistance.

Indeed we detect higher levels of FOXO target genes InR and 4E-BP in 9-(S)-HODE treated material (**Fig. 4F**), consistent with activation of FOXO after 9-(S)-HODE treatment. Work is ongoing to test the effects of prolonged exposure of whole animals to 9-(S)-HODE on metabolism using real-time NMR-based approaches but we feel that this work is best deployed as a logical extension of the current work. As detailed in minor point 1 below, concluding insulin resistance on the basis of FOXO activation without measuring alterations in metabolism and growth is speculative. We have accordingly abbreviated the title of the manuscript to "**Oxidised metabolites of the omega-6 fatty acid linoleic acid activate dFOXO**" to reflect the focus of the manuscript on FOXO localization and activity. Consideration that FOXO activation could confer insulin resistance is moved to the appropriate discussion section.

22. The authors discuss extensively about obesity and inflammation and insulin resistance. However, are larvae with hopTum obese? How is obesity related to the experiments discussed here? For example, the authors discuss about "Obesity coupled with low-grade inflammation is closely associated with the development of insulin resistance, and is speculated to require activation of stress-activated kinases like JNK". The two papers are related to work done in mice. Please add a paper supporting this statement in flies in connection to the mutant hopTum used here

We agree with the reviewer. The use of obese and obesity in the text is confusing and used too liberally. In the original draft we have used it as a (perhaps lazy) shorthand for changes in fatty acid flux/balance, which is actually what is investigated here and which lies at the heart of metabolic syndrome and the associated spectrum of co-occurring syndromes that include obesity, meta-inflammation and insulin resistance. Indeed, this was recognized in the first definition of metabolic syndrome by Grundy and co-workers (2005) (PMID: 15836891) where they state:

*“The metabolic syndrome is a common metabolic disorder that results from the increasing prevalence of obesity. The disorder is defined in various ways, but in the near future a new definition(s) will be applicable worldwide. The pathophysiology seems to be **largely attributable to insulin resistance with excessive flux of fatty acids implicated. A proinflammatory state probably contributes to the syndrome.**”*

This definition recognizes that it is the excessive flux of fatty acids (disturbance in levels and balance) that accompanies obesity that is the driver of insulin resistance and by definition control of the IIS pathway. In our experimental system we mimic this altered flux not by using obese flies (there is no evidence that hop[Tum] flies are obese) but rather by supplementing with defined families of FAs to alter the balance between FAs. We utilize the hop[Tum] background to generate a pro-inflammatory state that can then intersect with altered FA balance to determine whether this changes IIS signaling pathway which would be required for the development of insulin resistance.

To better reflect this we have stressed both in the Abstract, Introduction and Discussion that the key metric under investigation is changes in levels and balance of fatty acids. Obviously changes in levels and balance of fatty acids occur in obese states and would seem likely to be the driver of the complications associated with obesity, but in our system there is no evidence that flies are obese or need to be obese for changes in FA balance to have an effect.

Minor comments:

1. The title highlights parts of the study but it does not describe the main focus of the study. The current title appears to imply that the authors have characterized insulin signaling and function of Foxo, but the evidence is mainly on nuclear localization or increased nuclear stability and the expression profile in S2 cells.

This is a good point and in response to this and other comments made by the reviewer we have now abbreviated the title to “**Oxidised metabolites of the omega-6 fatty acid linoleic acid activate dFOXO**” to reflect the focus of the manuscript on FOXO localization and activity.

December 17, 2019

RE: Life Science Alliance Manuscript #LSA-2019-00356-TR

Dr. Paul W Badenhorst
University of Birmingham
Institute of Biomedical Research
Department of Anatomy
Edgbaston, West Mids B15 2TT
United Kingdom

Dear Dr. Badenhorst,

Thank you for submitting your revised manuscript entitled "Oxidised metabolites of the omega-6 fatty acid linoleic acid activate dFOXO". I have now assessed the data and changes introduced in revision and think that you are addressing the comments of the reviewers in an appropriate way. We would thus be happy to publish your paper in Life Science Alliance pending final small revisions, mainly to meet our formatting guidelines:

- I think it would have been good to include a mock control condition for figure 4 J, K to illustrate that the insulin treatment is working (response to rev#2, point 3); please include should you have this data at hand
- Please link your profile in our submission system to your ORCID iD, you should have received an email with instructions on how to do so
- Please add scale bars to Fig. 1B, 4A, 4F, 4H (the scale bar isn't visible), 5F, 5J, S4A
- Please mention the statistical tests performed in the figure legends wherever mentioning the p-values
- Please add information on the replicates performed for the q-RT-PCR analysis (can be done in Methods section)
- Datasets S1 and S2 were not provided, please do
- Please check the legend to figure 1 - I think there are mistakes in the panel descriptors

A. FINAL FILES:

B. MANUSCRIPT ORGANIZATION AND FORMATTING:

Sincerely,

Andrea Leibfried, PhD
Executive Editor

Life Science Alliance
Meyerohofstr. 1
69117 Heidelberg, Germany
t +49 6221 8891 502
e a.leibfried@life-science-alliance.org
www.life-science-alliance.org

As requested in your correspondence of 17 December we have addressed the outstanding points as below:

- I think it would have been good to include a mock control condition for figure 4 J, K to illustrate that the insulin treatment is working (response to rev#2, point 3); please include should you have this data at hand

As discussed we have performed these experiments in the presence of FCS which causes FOXO to be cytoplasmic unless 9-HODE is added. We observed little difference between mock and Insulin-treated (without 9-HODE).

- Please link your profile in our submission system to your ORCID iD, you should have received an email with instructions on how to do so

ORCID iD has been linked.

- Please add scale bars to Fig. 1B, 4A, 4F, 4H (the scale bar isn't visible), 5F, 5J, S4A Fig. 1B, 4A, 4F, 4H DONE (the scale bar isn't visible), 5F DONE, 5J, S4A DONE

We have added scale bars to allow indicated figure parts. New Fig1, Fig4 and Fig5 have been uploaded

- Please mention the statistical tests performed in the figure legends wherever mentioning the p-values

We have added the necessary information to all Figure legend sections – “determined using student’s t-test”.

- Please add information on the replicates performed for the q-RT-PCR analysis (can be done in Methods section)

Replicate information has been added to the materials and methods section

- Datasets S1 and S2 were not provided, please do

These datasets have been uploaded as supplementary datasets.

- Please check the legend to figure 1 - I think there are mistakes in the panel descriptors

Indeed, Figure 1 legend has been corrected.

Corrections/Changes are indicated in a marked-up version of the manuscript that has been submitted alongside a clean version of the manuscript.

December 23, 2019

RE: Life Science Alliance Manuscript #LSA-2019-00356-TRR

Dr. Paul W Badenhorst
University of Birmingham
Institute of Cancer and Genomic Sciences
Vincent Drive
Edgbaston, West Mids B15 2TT
United Kingdom

Dear Dr. Badenhorst,

Thank you for submitting your Research Article entitled "Oxidised metabolites of the omega-6 fatty acid linoleic acid activate dFOXO.". It is a pleasure to let you know that your manuscript is now accepted for publication in Life Science Alliance. Congratulations on this interesting work.

DISTRIBUTION OF MATERIALS:

Again, congratulations on a very nice paper. I hope you found the review process to be constructive and are pleased with how the manuscript was handled editorially. We look forward to future exciting submissions from your lab.

Sincerely,
